

# How does the explicit treatment of convection alter the precipitation-soil hydrology interaction in the Holocene African humid period?

Leonore Jungandreas[1,2], Cathy Hohenegger[1], and Martin Claussen[1,3]

[1]Max Planck Institute for Meteorology, Bundesstraße 53, 20146 Hamburg
[2]now affilated at German Centre for Integrative Biodiversity Research (iDiv) Halle-Jena-Leipzig
[3]Center for Earth System Research and Sustainability, Universität Hamburg, Bundesstraße 53, 20146 Hamburg

**Correspondence:** Leonore Jungandreas (leonore.jungandreas@mpimet.mpg.de)

**Abstract.** Global climate models with coarse horizontal resolution are largely unable to reproduce the monsoonal precipitation pattern over North Africa during the mid-Holocene. Here we present the first regional, storm-resolving simulations with an idealized but reasonable mid-Holocene vegetation cover. In these simulations, the West African monsoon expand farther north by about 4-5° and the precipitation gradient between the Guinea coast and the Sahara decreases in comparison to simulations
with a barren Sahara as it is today. The northward shift of monsoonal precipitation is caused by land surface – atmosphere interaction, i.e. the coupling of soil moisture and precipitation as well as interactions of the land surface with the large-scale monsoon circulation (e.g. the African easterly jet).

We find a similar response of the monsoon circulation to an increase in vegetation cover in simulations with parameterized convection. Moreover, changes are even larger than in simulations with explicitly resolved convection (i.e. the storm-resolving
simulations). We attribute the differences in monsoonal precipitation to differences in soil moisture that are strongly controlled by runoff and the precipitation characteristics as previously shown in Jungandreas et al. (2021).

We confirm this by performing simulations with a constant soil moisture field in both explicitly resolved and parameterized convection simulations. In these simulations, explicitly resolved convection simulations expand precipitation as far north as parameterized convection simulations. This study thus highlights the importance of the type of rainfall in modulating land-
atmosphere feedbacks, instead of only considering the amount of rainfall. Moreover, this study suggests that comprehensive land-surface schemes, which properly respond to varying precipitation characteristics, are needed for studying land-surface – atmosphere interaction.





## 1 Introduction

During the middle Holocene, about 5000 to 11000 years ago, the climate of the North African continent was wetter than today
and its landscape was characterized by a denser vegetation cover consisting of grass- and shrublands (Jolly et al., 1998) and
shaped by abundant lakes and wetlands (Tierney et al., 2017). This so-called "African humid period" (AHP) was triggered
by steady variations in the Earth's orbit (Kutzbach and Guetter, 1986; Street-Perrott et al., 1990). These variations lead to a
stronger northern hemispheric insolation gradient and thereby to an intensified West African Monsoon circulation (Kutzbach
and Otto-Bliesner, 1982; Kutzbach and Liu, 1997). It is widely accepted that the initial changes in the West African mon-
soon circulation were amplified by ocean-atmosphere (Kutzbach and Liu, 1997) and land-atmosphere feedbacks (Claussen and
Gayler, 1997; Braconnot et al., 1999, 2012; Krinner et al., 2012; Gaetani et al., 2017; Claussen et al., 2017), causing strong
changes in the hydrological cycle and vegetation, as indicated for example in sediment records or fossils in numerous proxy
data (e.g. Peyron et al. (2006); Bartlein et al. (2011)).

The intensified West African monsoon shifted monsoonal precipitation further north and led to an increase in soil moisture,
and hence, increased vegetation cover over the semi-arid and arid regions of the Sahel-Sahara region. In the simplest feedback,
increased soil moisture and enhanced vegetation cover increased evapotranspiration which in turn increased latent heat flux
and therefore lower-tropospheric humidity (e.g.Kutzbach et al. (1996); Doherty et al. (2000)). This favoured convective activity
and a positive feed back on precipitation.

Moreover, wetter soils and a higher vegetation cover are darker than bare soil, hence the surface albedo is lower (e.g.
Kutzbach and Liu (1997); Claussen and Gayler (1997); Texier et al. (1997); Broström et al. (1998); Doherty et al. (2000);
Braconnot et al. (2000); Vamborg et al. (2011)). A decreased surface albedo increased the available net surface radiation over
mid-Holocene North Africa. Levis et al. (2004) argued that the increase in net surface energy has raised the temperature over
the continent and enhanced the land-sea temperature gradient. This albedo-temperature feedback has dominated the direct soil
moisture-precipitation feedback (as explained above) and has thereby intensified the monsoon circulation and precipitation
over mid-Holocene North Africa. However, the temperature over North Africa can also decrease in response to a decrease
in albedo and an increase in surface net energy as found by Ripley et al. (1976) and Claussen and Gayler (1997). Due to
the higher moisture availability, most of the surplus of surface energy is transferred to the atmosphere via latent heat flux
rather than sensible heat flux. The decrease in the Bowen ratio (sensible heat flux divided by latent heat flux) increases the
vertical gradient of moist static energy within the boundary layer that destabilizes the atmosphere and favors convection and
precipitation (Schär et al., 1999).

Moreover, the decrease in temperature over North Africa (via enhanced latent heat flux) can affect the meridional tem-
perature gradient over North Africa and induce a dynamic land-atmosphere feedback as pointed out by Patricola and Cook
(2007) and Rachmayani et al. (2015). A weakened meridional temperature gradient slows down the African easterly jet due
to the thermal wind balance (Cook, 1999; Wu et al., 2009). This slowdown of the African easterly jet includes two coupling
mechanisms to convection and precipitation. Firstly, it decreases the mid-level moisture export out of North African and there-



fore increases the available moisture for convection and precipitation (Cook, 1999). Secondly, with the northward shift of the maximum temperature gradient, the African easterly jet core also shifts northward. This includes a broadening of the region of deep ascending motion between the African easterly jet and the Tropical easterly jet, supporting the broader occurrence of

convection and precipitation (Xue and Shukla, 1993, 1996; Cook, 1999; Nicholson and Grist, 2001; Grist and Nicholson, 2001).

However, in all paleo-climate studies, climate models still struggle to reproduce the precipitation distribution (i.e the amplitude and/or the extension of the tropical rainbelt) to support the greening of the semi-arid and arid regions of the North African continent (e.g. Joussaume et al. (1999); Braconnot et al. (2012); Harrison et al. (2015); Brierley et al. (2020)), as compared

to proxy data. A possible explanation for diverging rainfall distributions could be the parameterization of convective rainfall, which is used in coarse-resolution climate models (e.g. Yang and Slingo (2001); Randall et al. (2003); Stephens et al. (2010); Dirmeyer et al. (2012); Fiedler et al. (2020); Jungandreas et al. (2021)). The influence of the representation of convection on the mid-Holocene West African monsoon but with present-day land surface cover was investigated by Jungandreas et al. (2021). Although, they found no substantial effect on the meridional precipitation distribution over mid-Holocene North Africa when

comparing simulations with parameterized and explicitly resolved convection, they identified an important feedback between the land surface and precipitation that depends on how convection is represented in the model. Resolving convection explicitly leads to less frequent but more local and intense precipitation. In contrast, parameterizing convection causes less intense but more frequent precipitation that steadily moistens the soil by generating less runoff. As the soil is not able to absorb the high amount of rainfall in explicitly resolved convection simulations, a large fraction is removed from the system as runoff and thus

causes a noticeably weaker refilling of the soil moisture during the monsoon season as compared to parameterized convection simulations. Consequently, in explicitly resolved convection simulations, the lower soil moisture yields a lower latent heat flux and consequently an extenuated convective activity, while in parameterized convection simulations the higher latent heat flux further supports convection and maintains a strong positive precipitation feedback.

In their simulation, Jungandreas et al. (2021) have prescribed present-day conditions for the vegetation cover, like the sim-

ulation setup used in the Paleoclimate Modeling Intercomparison Project phase 1 (PMIP1). This in relation to early and mid-Holocene conditions unrealistic specification of land-surface conditions prompts the question of whether more realistic land-surface coverage and thus, higher soil moisture and modified hydrological conditions, affect the precipitation distribution in explicitly resolved convection simulations as compared to parameterized convection simulations.

In the present study, we first examine the most important land-atmosphere feedbacks in our storm-resolving simulations that

evolve due to a higher vegetation cover (Sec. 3.1). To do so, we prescribe a larger, mid-Holocene like vegetation cover over North Africa based on the simulated total vegetation cover fraction of the transient mid-Holocene simulations of the Max-Planck Institute Earth System Model (MPI-ESM) (Dallmeyer et al., 2020). The identified feedbacks are qualitatively similar in parameterized convection simulations (Sec. 3.2). However, despite the same prescribed vegetation cover, soil moisture strongly controls the hydrological cycle and the precipitation response in explicitly resolved convection simulations compared

to parameterized convection simulations. We confirm the strong dependence of the hydrological cycle on the representation of





convection by performing simulations with the same constant soil moisture in explicitly resolved and parameterized convection simulations (Sec. 3.3).

## 2 Methods

### 2.1 Model

The model and simulation setup used in this study is identical to the one used in Jungandreas et al. (2021). We use the ICON (ICOsahedral Nonhydrostatic) model framework version 2.5.0 (Zängl et al., 2015) in its operational Numerical Weather Prediction (NWP) mode. The ICON-NWP model framework supports nested experiments. The nesting allows us to run coarse horizontal resolution simulations with parameterized convection and high-resolution simulations using explicitly resolved convection, simultaneously. We simulate using a one-way nesting strategy. The convective parametrization used in our simulations

is based on the bulk mass-flux approach introduced by Tiedtke (1989) with modifications by Bechtold et al. (2014). Zängl et al. (2015) lists the other physical parameterizations of the model framework. All simulations are limited-area simulations. The domains are outlined in Fig. 1 as well as the main analysis domain we use in our investigations.

### 2.2 Simulation Setup

We perform a 30-year spinup simulation that covers the period from 7039 y b2k (years before the year 2000) to 7010 y B2k.

The spinup simulation is conducted on a regional domain with a 40 km horizontal grid spacing and 75 vertical levels (blue domain outlined in Fig. 1) using parameterized convection. The soil moisture reaches a stable state after around 15 years (of the 30-year spinup period). We select two years after this 15-year soil moisture-spinup phase and start our nesting experiments for the boreal summer monsoon season. The nesting experiments are initialized on May 30 and run for five months (JJASO). The initial and boundary data originate from the transient, global Holocene simulation conducted with the MPI-Earth System

Model (ESM). More details about the transient MPI-ESM Holocene simulation are provided by Dallmeyer et al. (2020).

The parent domain of the nested simulation is identical to the domain of the spinup simulation with the same horizontal and vertical resolution. Inside the 40km domain, further nested domains are embedded with horizontal resolutions of 20 km, 10 km, and 5 km (Fig. 1). While the simulations with 40 km, 20 km, and 10 km grid spacing use parameterized convection, the 5 km domain uses explicitly resolved convection. In the following, we refer to the simulations with 40 km horizontal grid spacing and

parameterized convection as 40km-P simulations. Similarly, the 5km resolution simulation with explicitly resolved convection is labeled with 5km-E. The nested simulations are initialized one hour after another by their respective parent domain. Lateral boundary conditions for the nested simulations are also obtained from their parent simulation and updated every 6 hours. We further prescribe 6-hourly sea surface temperature (SST) and sea ice (SIC) fields from the transient MPI-ESM Holocene simulation, as well as the same orbital parameters and tracer gases carbon dioxide ($CO_2$), methane ($CH_4$) and nitrogen oxide

($N_2O$).





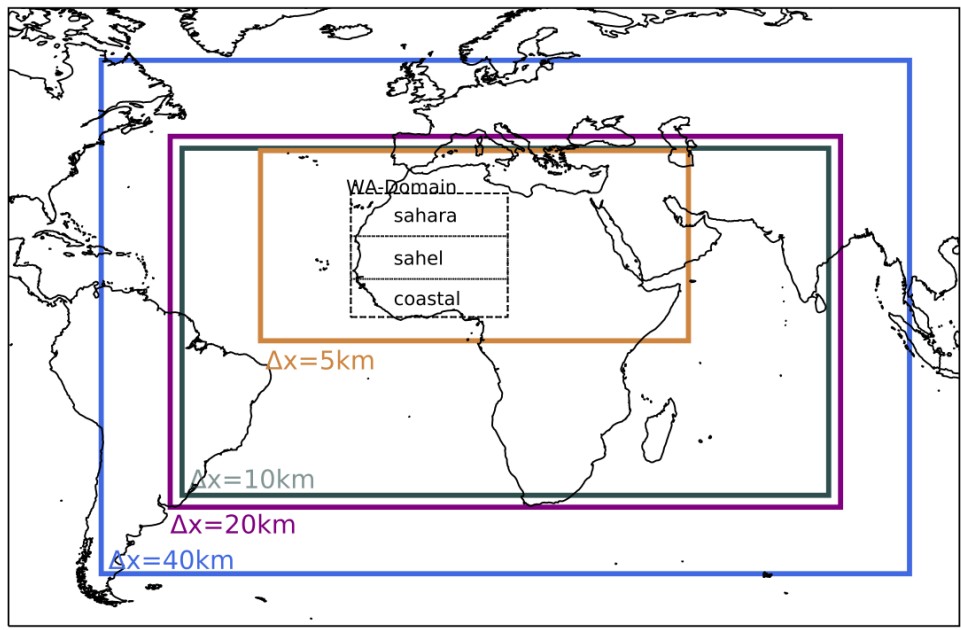

**Figure 1.** Solid, colored domains outline the nesting domains of the simulations for the various grid spacings (as indicated). The dashed, black domain labeled with "WA-Domain" (West Africa) displays the main analysis domain. The WA-Domain spans the area from 5 °N-31 °N and from 18 °E-15 °W. The three dotted, black domains within the WA-Domain are used to distinguish the coastal African region ("coastal"), the Sahel region ("Sahel"), and the Saharan region ("Sahara") that are indicated in the longitudinal-mean plots. The coastal region spans from 5 °N-13 °N and from 18 °E-15 °W. The Sahel region spans from 13 °N-22 °N and from 18 °E-15 °W and the Sahara region from 22 °N-31 °N and from 18 °E-15 °W.

In our analysis (Sec. 3), we focus on the three strongest monsoon months from July to September (JAS). Further, we investigate both, the 40 km parameterized convection (40km-P) and the 5 km explicitly resolved convection simulations (5km-E), to analyze whether and how differently the atmosphere responds to the underlying surface conditions in both representations of convection. To address the influence of the horizontal resolution on our results, we redo our analysis but compare the 10km-P and 10km-E simulations. This analysis confirms that the feedback mechanisms are similar to the 40km-P and 5km-E simulations, thus are mainly caused by the representation of convection rather than by the horizontal resolution. However, differences between simulations with parameterized and explicit convection are not as pronounced, because 10km horizontal resolution already allows for partly explicit calculations of convective processes even in simulations with parameterized convection (Appendix B).

### 2.2.1 The "Green Sahara" simulations (GS)

To investigate the influence of the land surface on the monsoonal rainbelt in parameterized and explicitly resolved convection simulations, we compare simulations with present-day vegetation cover (Jungandreas et al. (2021), Fig.2a and c) to simulations





with a higher, more realistic mid-Holocene-like vegetation cover (Fig.2b and d). Because the largest area of North Africa in the simulations with present-day vegetation cover is bare soil (desert) or sparse vegetation we refer to these simulations in

the following as "DS"-simulations (for "Desert Sahara"). The land-surface cover of these simulations are obtained from the Integrated Weather Forecast System (IFS). The simulations with a higher vegetation cover are introduced below.

We prescribe an idealized, denser vegetation cover over the whole 40km simulation domain (spanning from 70.5°W - 99.5°E and from 59°N to 49°S) guided by the MPI-ESM Holocene simulations (see Dallmeyer et al. (2020)). Based on the simulated desert fraction (see Fig. A1 in Appendix A) and/or the preset present-day vegetation type, we extend the area of evergreen

tropical rainforest (only determined by the present-day vegetation type) over North Africa (Fig. 2b, dark green) to about 15°N. North of the rainforest, we prescribe a decreasing vegetation gradient from closed to open shrubland (light yellow, desert fraction <0.2 and depending on the present-day vegetation type), to closed to open herbaceous vegetation (dark red, desert fraction between 0.2 - 0.6 and if the present-day vegetation type is "bare area"), to sparse vegetation (lighter red, desert fraction between 0.6 - 0.9 and if the present-day vegetation type is "bare area"). Bare soil (desert) prevails only over a small area over

Egypt (desert fraction >0.9). Table A1 gives the exact ranges and values of the prescribed vegetation types. In the following, we label these vegetated Sahara simulations with "GS" (for "Green Sahara").

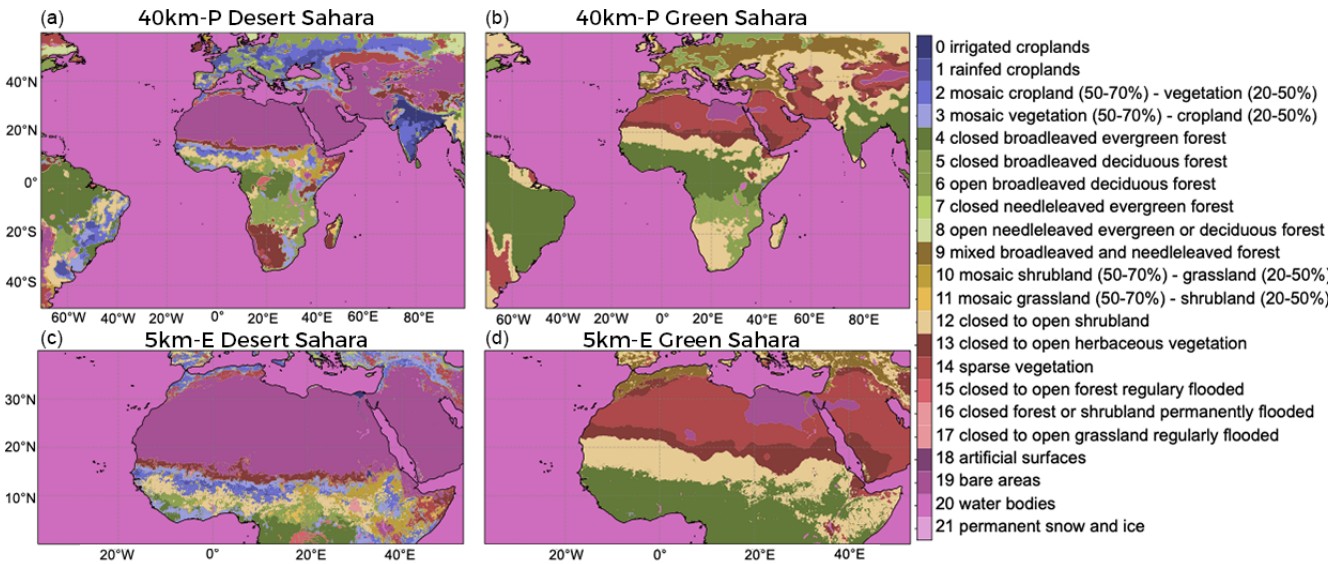

**Figure 2.** Land surface cover for the "Desert Sahara" (a,c) and "Green Sahara" (b,d) simulations for the 40 km and the 5 km simulations, respectively.

With the vegetation cover we adjust all variables that depend on the vegetation type for example the leaf area index, surface albedo, root depth, minimal stomata resistance, and several others (see Table A2). Based on the present-day values, which




originate from the ERA-reanalysis dataset, we prescribe spatially constant values and adjust them to be consistent for the
respective vegetation type. We calculated these spatially constant values as follows:

    1. We calculate the dominant land cover type ($\equiv$ vegetation type) for each grid cell of the present-day land cover distribution of the 40km-domain (see Fig. 2a).

    2. All grid cells with the same dominant vegetation type are used to calculate the mean and either the 75th or the 85th percentile value for each variable.

3. Ee prescribe these mean or percentile values to the idealized mid-Holocene land-cover types, respectively.

    ICON-NWP by default only calculates with one vegetation type per grid box, the dominant one, and not with fractions of several vegetation types as preset by the IFS dataset. The surface variables calculated and preset by the IFS, for example, albedo or leaf area index, therefore, consider the presence of different vegetation types per grid box. By using only the dominant vegetation type for the calculations of the constant surface variables, the mean value sometimes underestimates the variable
values as compared to the present-day values. In these cases, we use the 75th or the 85th percentile values. Table A2 in Appendix A summarizes all modified variables and indicates if we use mean or percentile values.

### 2.2.2 GS simulations with constant soil moisture GS-cSM

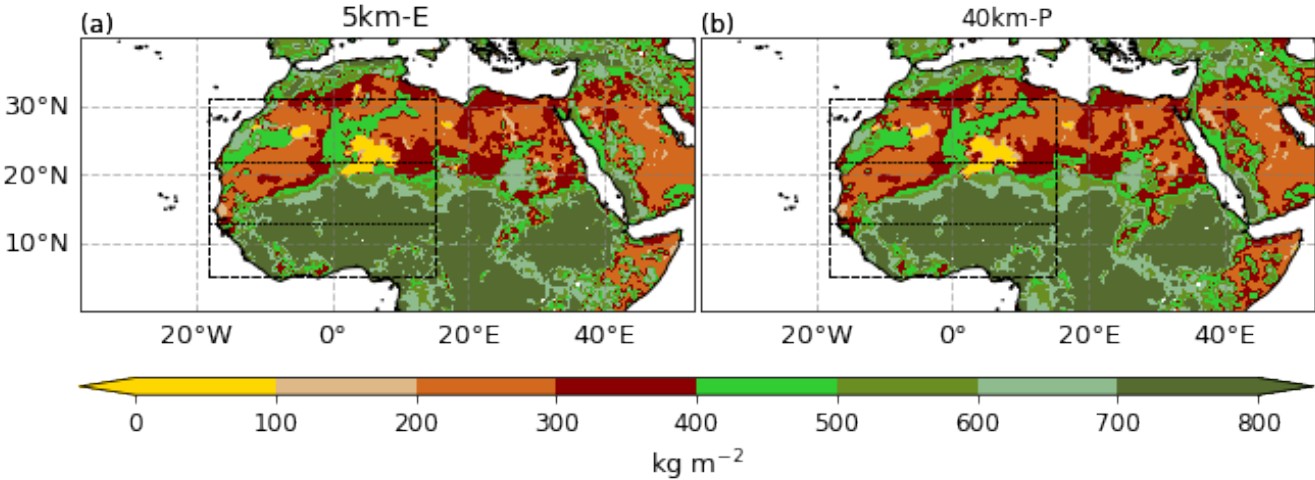

**Figure 3.** The prescribed constant soil moisture field as sum over the uppermost six soil layers (up to a depth of 3.42 m) in the 5km-E (a) and the 40km-P (b) simulation. The vertical black, dashed lines indicate the borders of the coastal, the Sahel, and the Sahara region outlined in Fig. 1.

    In addition to the GS simulations, we perform a second set of GS simulations, but with a prescribed soil moisture field that we keep constant for the whole simulation period (Fig. 3). We label these simulations with GS-cSM (for "Green Sahara with



constant soil moisture"). We prescribe the soil moisture field of the 1st September 00 UTC of the 40km-P GS simulation to all four domains at all times. The soil moisture field from the 1st September displays relatively high soil moisture levels, compared to the soil moisture levels at the beginning of the GS-simulations, to provide enough soil moisture for evapotranspiration. Little variations between the 5km-E and 40km-P soil moisture field are possible due to interpolation. Apart from keeping the soil moisture constant, the simulation setup in the GS-cSM simulations is identical to the GS simulation setup.

## 3    Results and Discussion

In the following section (Sec. 3.1), we compare the storm-resolving simulations with present-day land surface cover (DS) introduced by Jungandreas et al. (2021) with the simulations with higher vegetation cover (GS, Sec. 2.2.1) to describe the evolving land-atmosphere feedbacks. Subsequently, in Sec. 3.2 we briefly highlight the differences between parameterized (40km-P) and explicitly resolved (5km-E) convection simulations. In Sec 3.3 we present the results of our sensitivity experiment

in which we prescribe the same constant soil moisture field in the 5km-E and the 40km-P simulations.

All analyses are done for the months of July to September over the WA-domain and, if not further specified, for the subregions "coastal", "Sahel" or "Sahara" as shown in Fig. 1.

### 3.1    Land-atmosphere coupling - how the land influences the large-scale monsoon circulation

#### 3.1.1    Changes in vegetation and the surface energy budget

In our simulations, the main influence of the prescribed change in vegetation cover is, on the one hand, on the energy partitioning into latent and sensible heat flux and, on the other hand, on the temperature gradient over the North African continent between the drier, warmer region of the Sahara and the cooler, moister regions near the coast.

In the GS simulation, we prescribe a higher vegetation cover (indicated by the Normalized Difference vegetation index (NDVI), Fig. 4 a) over the whole WA-Domain, with the strongest increase over the Sahel region compared to the DS simulation.

Prescribing a higher vegetation cover strongly decreases the surface albedo by about 54% (compare energy fluxes in Table 1). The decrease in albedo implies about 50% less reflection of solar incoming radiation at the surface, thus higher absorption of solar radiation. This remains true despite the decrease in incoming solar radiation due to higher cloud cover. The thermal outgoing radiation at the surface slightly decreases due to mostly colder temperature and the longwave, downward radiation increases due to higher cloud cover. Taking these two effects together, the surface gains more thermal energy. All in all, net

radiation increases and the total heat flux (sensible + latent heat flux) from the surface into the atmosphere increases. How much of the total energy is transformed into latent and sensible heat strongly depends on the water availability at the surface.

Over the coastal region, moisture is generally not limited (regardless of the prevailing vegetation type i.e. dense shrubland or forest). Hence, latent heat flux is high and sensible heat flux small in both, the GS and DS simulation (Fig.4 b and c). Therefore, variations between the GS and the DS simulations are small. In contrast, over the Sahel (a region strongly controlled by water

availability) the higher vegetation cover increases the interception storage of water and makes deeper soil water available for





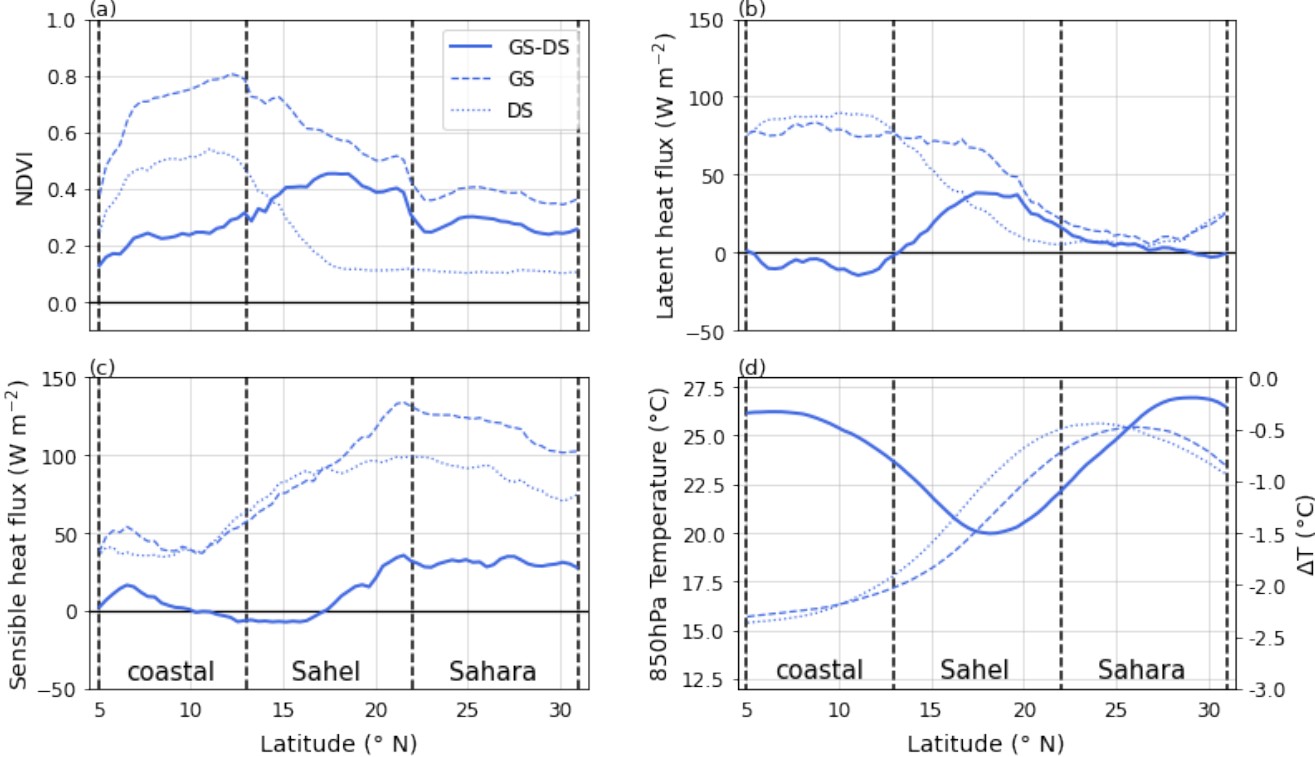

**Figure 4.** JAS-mean meridional distribution of the normalized difference vegetation index (NDVI; a), the latent heat flux (b), the sensible heat flux (c) and the 850hPa-temperature (d) for the 5km explicitly resolved convection simulations. Note the second y-axis in the panel d for the difference temperature (solid line). The longitudinal mean is taken over the WA-Domain for the 5km-E simulations. The lines indicate the distribution for the Desert Sahara-simulation (DS, blue dotted line), for the Green Sahara-simulation (GS, blue dashed line), and the difference between the GS and the DS simulation (blue solid line). The vertical black, dashed lines indicate the borders of the coastal, the Sahel, and the Sahara region outlined in Fig. 1.

evapotranspiration. Therefore, the total increase of the surface heat fluxes (compare Fig. 4 b and c) over the Sahel is dominated by the increase in latent heat flux. Over the Sahara, water remains too limited and the increase in sensible heat flux dominates the total increase in the surface heat fluxes in the GS as compared to the DS simulation.

Differences in the temperature field are dominantly determined by differences in the latent heat flux. Latent heat flux is more
effective in transporting energy from the surface into the atmosphere and can cool the surface. As shown in Fig. 4 d, lower-tropospheric temperature (here shown for 850hPa) decreases in the GS compared to the DS simulation. The strongest decrease in temperature of about 1.5° C is located over the Sahel region and coincides with the region of the strongest increase in latent heat flux. The maximum temperature over the Sahara shifts northward where vegetation cover is less dense and thus, latent heat flux is smaller. Simultaneously, the temperature over the coastal region does not change substantially, consistent with the
small variations in latent heat flux. Consequently and as indicated by the difference in temperature (solid line, note the second



| | 5 km-E DS | 5 km-E GS |
|---|---|---|
| SW↓ (W m$^{-2}$) | 280.52 | 253.58 |
| SW↑ (W m$^{-2}$) | 67.87 | 34.15 |
| SW$_{net}$ (W m$^{-2}$) | 212.65 | 219.43 |
| LW↓ (W m$^{-2}$) | 393.06 | 402.82 |
| LW↑ (W m$^{-2}$) | 476.29 | 470.84 |
| LW$_{net}$ (W m$^{-2}$) | -83.22 | -68.01 |
| R$_{net}$ (W m$^{-2}$) | 129.43 | 151.42 |
| LH (W m$^{-2}$) | 41.03 | 48.57 |
| SH (W m$^{-2}$) | 72.81 | 87.86 |
| Res (W m$^{-2}$) | 15.59 | 15.56 |

**Table 1.** JAS-mean values of radiation components at the surface averaged over the WA-Domain (Fig. 1) for the 5 km-E DS and GS simulations. The abbreviations used are as follows: SW↓ - downward, shortwave radiation, SW↑ - upward, shortwave radiation, SW$_{net}$ - net shortwave radiation, LW↓ - downward, longwave radiation, LW↑ - upward, longwave radiation, LW$_{net}$ - net longwave radiation, R$_{net}$ - net radiation, LH - latent heat flux, SH - sensible heat flux, Res - Residuum term.

y-axis), the temperature gradient south of about 18°N (location of the maximum decrease) weakens, while it strengthens north of 18°N. The maximum temperature gradient shifts north by about 2-3° in the GS compared to the DS simulation (not shown). This temperature coupling is especially important for the formation and the location of the African easterly jet over North Africa thus influences the dynamics of the atmosphere and will be further discussed in Section 3.1.2.

**3.1.2 Changes in atmospheric dynamics**

Despite the decrease in temperature (Fig. 4 d), the low pressure system (not shown) over the Sahara deepens. This strengthening of the low pressure system is caused by a stronger thermal uplift at the surface which in turn is caused by the increase in net radiation (SW+LW; see Table. 1) at the surface over the northern Sahel and Sahara region. Therefore, the surface pressure gradient between the Sahara and the tropical Atlantic Ocean increases and drives a stronger low-level, southwesterly monsoon

flow (Fig. 5). The strengthened southwesterly winds transport cool and moist air from the Gulf of Guinea and the coastal region (consistent with the drying of the coastal region Fig. 7) deep into the African continent and provide additional moisture for convection and precipitation. Moreover, in response to the strengthened low-level, southwesterly winds, the location of the Innertropical Front (ITF, the location where the southwesterly monsoon flow converges with the northeasterly Harmattan winds) shifts about 2° further north (from about 20°N to 22°N; Fig. 5). The low and mid-level lifting over the northern Sahel

and southern Sahara region that is associated with the lifting of air at the ITF (between 20-23°N) (Nicholson, 2009; Thorncroft et al., 2011) slightly strengthens and reaches higher altitudes in the GS simulation (Fig. 6 c).





The decrease of the temperature gradient south of about 18°N (Fig. 4 d) and the northward shift of the maximum temperature gradient implies a weakening of the meridional gradient in geopotential. The weakening of the geopotential gradient weakens the African easterly jet (AEJ; indicated by the local maximum of easterly wind speed (blueish color) at about 600hPa) and
shifts its core about 3° north in the GS compared to the DS simulation (Fig. 5; Cook (1999); Wu et al. (2009)). Several studies (e.g. Cook (1999); Nicholson and Grist (2001); Grist and Nicholson (2001)) confirm that the northward displacement and a weakening of the AEJ are associated with more humid conditions over the Sahel. Hence, more moisture remains over North Africa as a source for clouds and precipitation in the GS simulations.

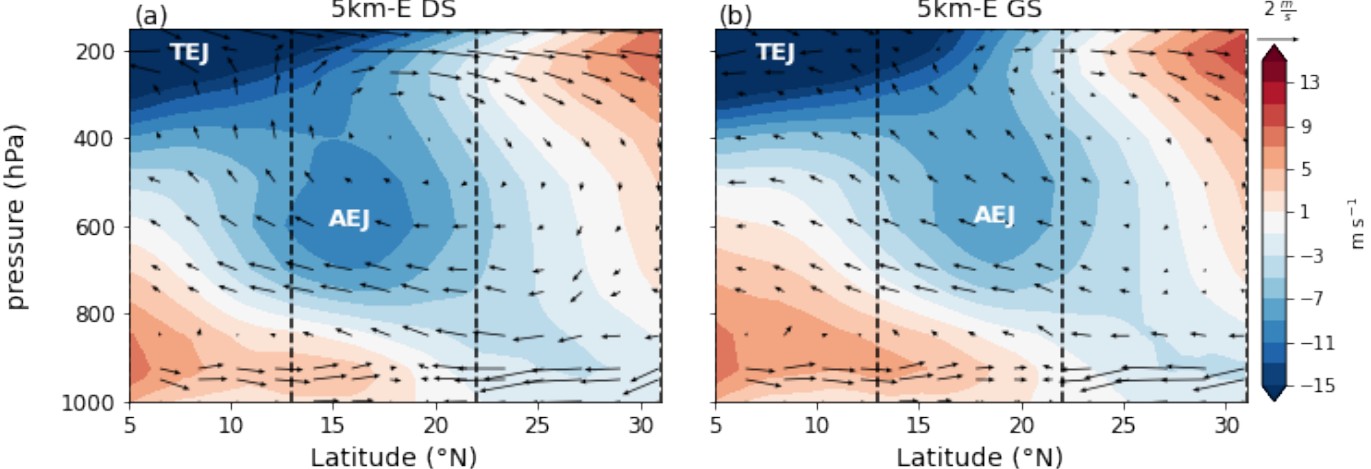

**Figure 5.** Vertical cross-section of the wind field over the WA-domain (Fig. 1). The shading displays the mean zonal wind, and the arrows the mean meridional and vertical wind for the 5km-E and the DS (a) and the GS (b) simulation, respectively. Note that the vertical wind speed is multiplied by 100 to make arrows better visible. The location of the African easterly jet (AEJ) and the Tropical easterly jet (TEJ) are indicated by the labels. The dashed vertical lines indicate the borders of the coastal, Sahel, and Sahara domain also displayed in Fig. 1.

Moreover, with the northward shift of the AEJ, the region of strong ascent between the Tropical easterly jet (TEJ; maximum
easterly winds at about 200hPa) and the AEJ axes (e.g.Grist and Nicholson (2001); Nicholson and Grist (2001)) broadens and reaches further north. Fig. 6 c displays a decrease in vertical upward motion at mid- and high-levels over the coastal region and an increase over the Sahel-Sahara region up to about 25°N in the GS compared to the DS simulation. The ascending motion between the jet axes is part of a deep meridional overturning circulation (Thorncroft et al., 2011). It not only distributes moisture within the whole troposphere (Fig. 7), but it also forces the low-level inflow of fresh, moist monsoonal air from the
south and southwest. Hence, it potentially contributes to the strengthening of the low-level, southwesterly monsoon winds in the GS simulations. Moreover, stronger convection in turn yields higher latent heat release, enhances the temperature gradient, and reinforces the monsoon circulation. Stronger upward motion and higher atmospheric humidity lead to more supportive conditions for convection and therefore support higher precipitation rates over the Sahel and Sahara.



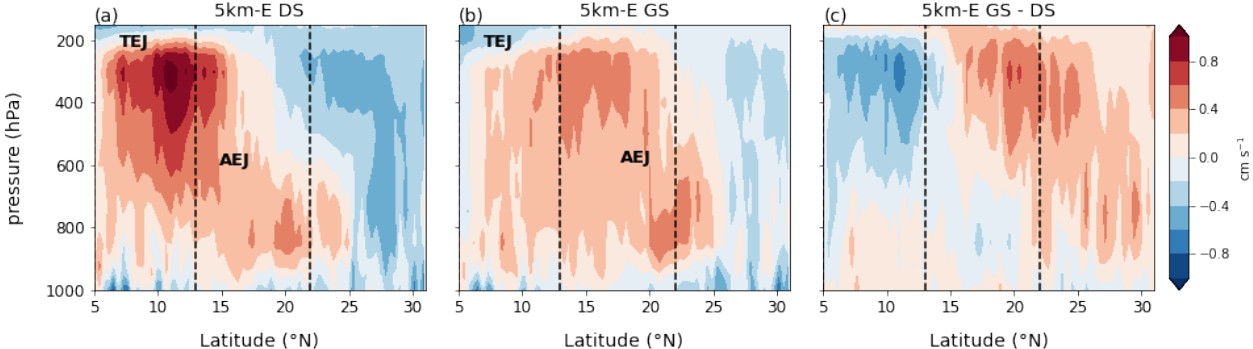

**Figure 6.** Vertical cross section of the vertical wind component for the 5km-E DS (a) and GS (b), and for the difference between the GS and DS (c) simulation, respectively, averaged over the WA-Domain outlined in Fig. 1. Again, the location of the African easterly jet (AEJ) and the Tropical easterly jet (TEJ) are indicated by the labels. The dashed vertical lines again indicate the latitudes of the coastal, Sahel and Sahara domain also displayed in Fig. 1.

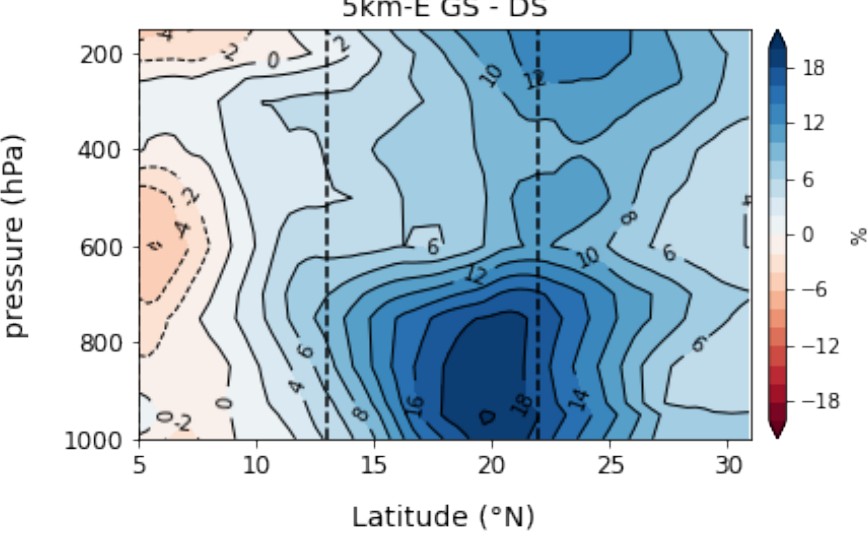

**Figure 7.** Vertical cross section of the difference in relative humidity between the GS and the DS simulation for the 5 km-E simulations. The field is averaged over the WA-Domain outlined in Fig. 1. The vertical dashed lines delineate the latitudes of the coastal, the Sahel and the Sahara domain (see Fig. 1).

### 3.1.3 Changes in atmospheric thermodynamics

The change in surface latent heat flux together with the strengthened monsoon circulation influences the moisture in the lower troposphere as shown in Fig. 7. Over the coastal region relative humidity decreases slightly in the GS compared to the DS simulation. The decrease in latent heat flux, and the intensified southwesterly monsoon winds yield a higher moisture divergence over the coastal region. In contrast, the increase in relative humidity in the lowest atmosphere layers over the Sahel and Sahara region can be linked to the increase in latent heat flux (Fig. 4 b), the decrease in temperature (Fig. 4 d) and the

convergence of moisture (not shown). This increase in boundary-layer relative humidity supports the triggering of convection





over these regions. In turn, the increase in convective activity likely contributes to the increase in the upper tropospheric humidity as it distributes boundary layer-moisture upwards.

Changes in the stability parameters, such as CAPE (Convective Available Potential Energy) or CIN (Convective Inhibition) listed in Table 2, are consistent with the less supportive conditions for the triggering of convection in the GS simulation over
the coastal region and the more supportive conditions over the Sahel and Sahara region. Over the coastal region, CAPE slightly decreases and, despite the CIN becomes slightly less negative, the LFC is higher. Over the Sahel region, CAPE increases, CIN becomes less negative and the LFC lowers. Changes over the Sahara are very weak and indicate generally poor conditions for convection and precipitation to occur.

|  |  | 5 km-E DS | 5 km-E GS |
|---|---|---|---|
| coastal | CAPE (J kg$^{-1}$) | 706.7 | 505.6 |
|  | CIN (J kg$^{-1}$) | -37.5 | -34.1 |
|  | LFC (m) | 1131 | 1236 |
|  | cloud cover (%) | 77 | 79 |
| sahel | CAPE (J kg$^{-1}$) | 159.8 | 220.6 |
|  | CIN (J kg$^{-1}$) | -262.0 | -211.5 |
|  | LFC (m) | 2617 | 1940 |
|  | cloud cover (%) | 36 | 58 |
| sahara | CAPE (J kg$^{-1}$) | 0.0 | 2.1 |
|  | CIN (J kg$^{-1}$) | 0.0 | -102.9 |
|  | LFC (m) | 3321 | 2974 |
|  | cloud cover (%) | 20 | 36 |

**Table 2.** JAS-mean values of 12 UTC CAPE and CIN and JAS-mean level of free convection (LFC) and total cloud cover for the coastal, Sahel and Sahara region (Fig. 1) for the 5 km-E DS and GS simulation.

### 3.1.4 Resulting changes in precipitation

The land-atmosphere coupling described above induces a positive land-atmosphere feedback over the Sahel-Sahara region, where the mean precipitation increases in response to increased vegetation cover, and a negative feedback over the coastal region (Fig. 8; increased vegetation cover that leads to a decrease in precipitation which counteracts the growing of plants). Over the coastal region, the increase in vegetation cover leads to a decrease in latent heat flux (Sec. 3.1.1) and humidity, less favourable thermodynamic conditions (Sec. 3.1.3), and weakened vertical upward motion (Fig. 6). As a result, precipitation
decreases over the coastal region. Vice versa, over the Sahel region (and to a lesser extent over the Sahara region), the strong increase in latent heat flux and lower-tropospheric humidity, the more supportive thermodynamic conditions as well as the weakening and northward shift of the AEJ (Sec. 3.1.2) that enhances the monsoon circulation and atmospheric humidity, leads





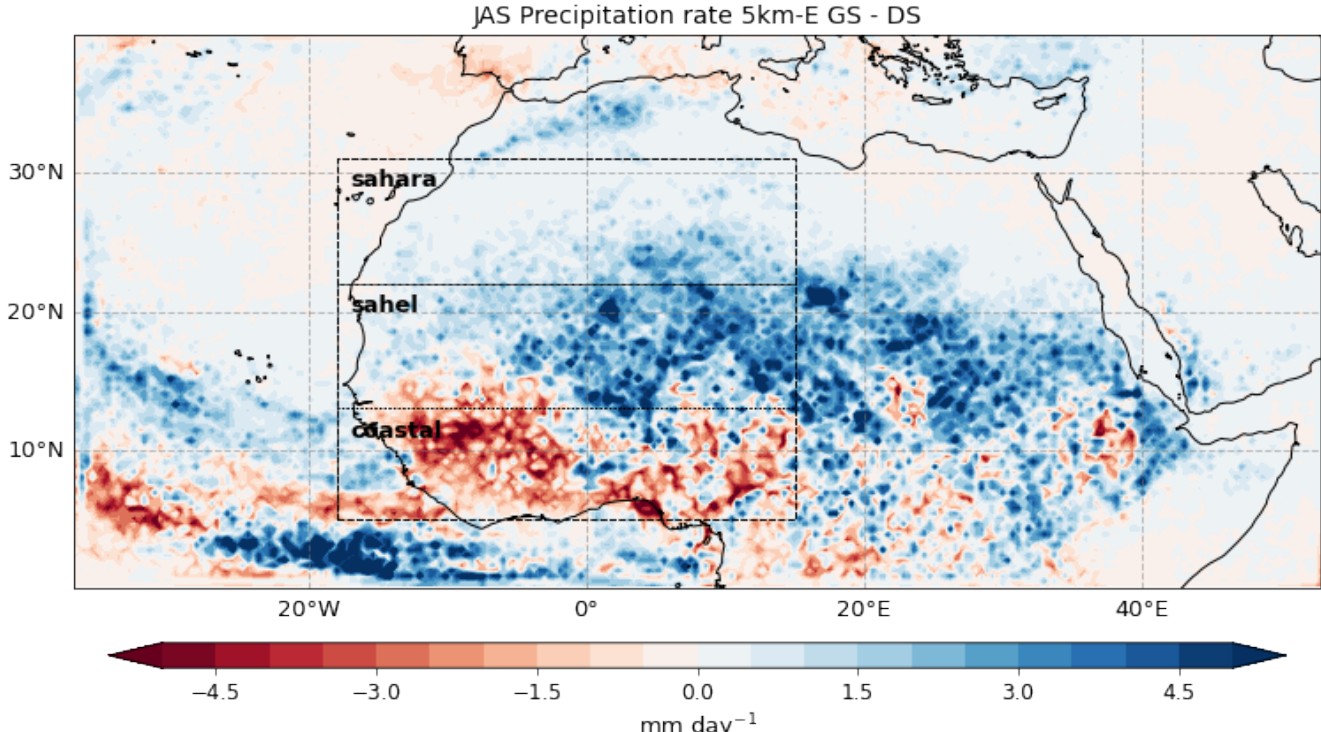

**Figure 8.** JAS-mean difference in precipitation rate for the between the 5km-E GS and DS simulation. The dashed box indicates the WA-analysis domain.

to a strong increase in precipitation. Overall, the rainbelt shifts further north by about 4-5° (from the coastal into the Sahel-Sahara region), rather than increasing uniformly over the whole WA-domain. Therefore, we suggest that the main cause for the

further northward propagation of monsoonal precipitation in the GS simulation is of dynamical nature. However, to quantify which feedback dominates the precipitation response, further analyses are needed and are beyond the scope of this paper.

## 3.2 Positive land-atmosphere feedbacks but with different strength - differences between 40km-P and 5km-E simulations

In the previous section, we have analyzed the differences between the 5km-E GS and DS simulation to identify the operating

land-atmosphere feedbacks in our storm-resolving simulations. All feedback mechanisms explained in Sec. 3.1 qualitatively also hold for simulations with parameterized convection and for all horizontal resolutions. Both representations of convection show a positive precipitation coupling over the Sahel-Sahara region, and a negative coupling over the coastal region, in response to an increased vegetation cover. Beside these similarities in the general feedback sign, there are pronounced differences in the magnitude of the changes between explicitly resolved and parameterized convection simulations.





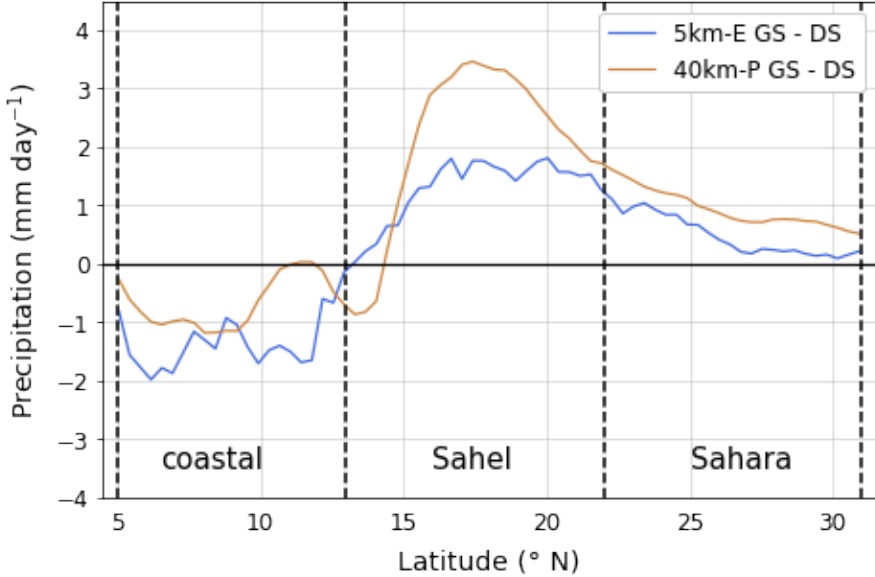

**Figure 9.** Difference of the JAS-mean-meridional precipitation distribution between the GS and DS simulation for the 5km-E simulations (blue line) and the 40km-P simulation (orange line), respectively. The vertical black, dashed lines indicate the borders of the coastal, the Sahel, and the Sahara region outlined in Fig. 1.

Fig. 9 displays strong differences in the precipitation response to changes in the land surface cover between both representations of convection. The decrease in precipitation in the 5km-E GS compared to the DS simulation over the coastal region is stronger while the increase over the Sahel and Sahara region is noticeably weaker than in the 40km-P simulations. Over the Sahara, the precipitation increase in the 40km-P simulations remains higher but differences to the 5km-E simulations become smaller. We find that precipitation in the 40km-P simulations is generally much higher throughout the WA-domain and extends farther north (by about $2°$ taking $2\,\mathrm{mm\,day^{-1}}$ as threshold) as compared to the 5km-E simulations (not shown, refer to Jungandreas et al. (2021) for the DS simulations).

We attribute the different precipitation signal to the precipitation-runoff-soil moisture mechanism described for the DS-simulations in Jungandreas et al. (2021). Precipitation in the 5km-E simulations is more local and intense, leading to much larger amounts of runoff relative to the precipitation amount (Fig. 10 a). In the mean over the WA-domain, about 37% and 35% of precipitation become runoff in the 5km-E GS and the DS simulation, respectively, while in the 40km-P GS and DS simulations only about 21% and 20% of the precipitation leaves the system as runoff, respectively. As a result of the higher runoff-precipitation ratio, the daily-change in soil moisture is much smaller in the 5km-E as compared to the 40km-P GS simulation (Fig. 10 b) (this is also true for the DS simulations). This implies that much less of the surplus of precipitation in the 5km-E GS simulation (compared to the DS simulation) is stored in the soil (especially over the Sahel and Sahara region) as compared to the 40km-E simulations, resulting in generally lower total soil moisture content in the explicitly resolved convection simulations (not shown).

We argue that the precipitation-runoff-soil moisture mechanism dampens the potential precipitation response to a vegetated Sahel-Sahara region in the 5km-E compared to the 40km-P simulations. A weaker precipitation response in the 5km-E simulation could result both from a weaker response of precipitation to a given change in latent heat flux, as for instance argued



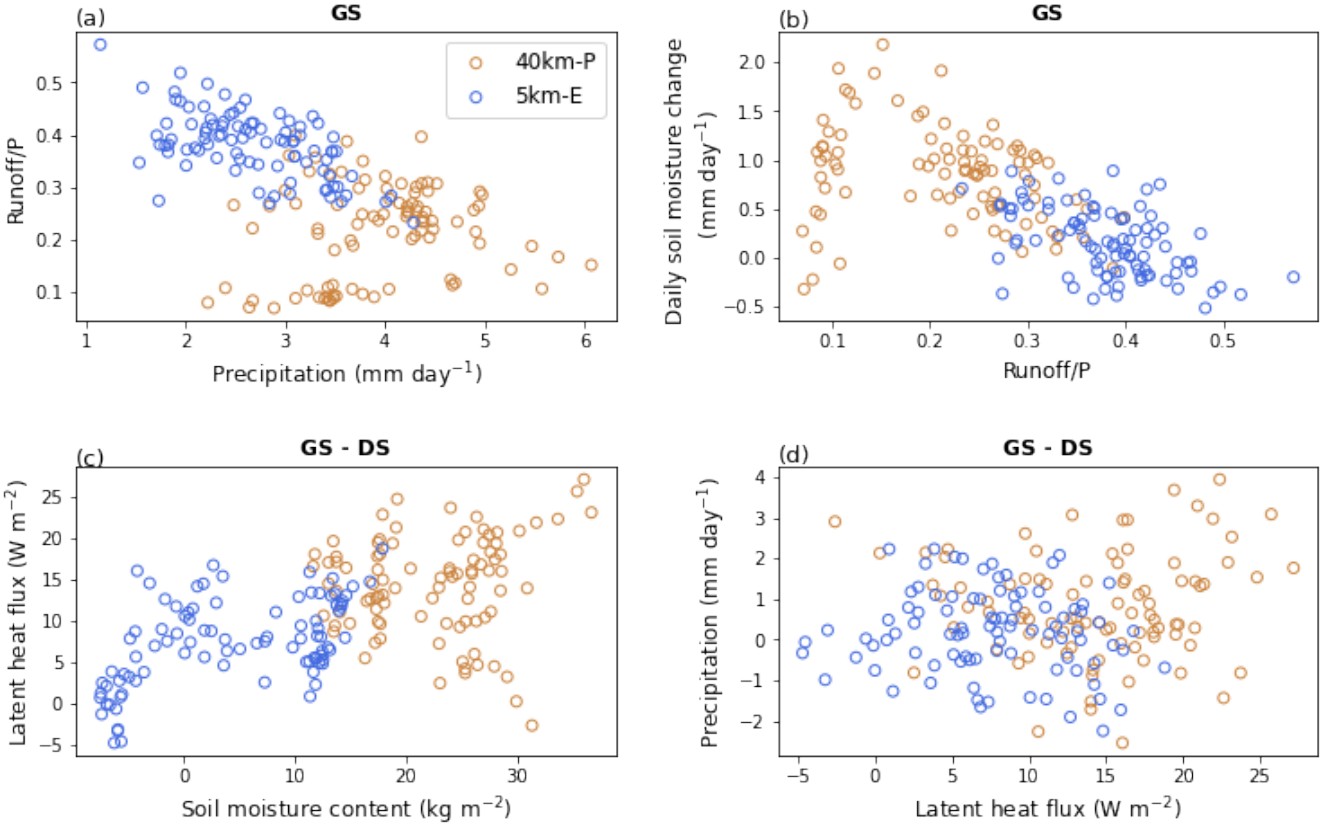

**Figure 10.** Relationship between different variables in the GS simulation in Panel a and b, and the difference between the GS and the DS simulation in Panel c and d for the 5km-E simulations (blue circles) and the 40km-P simulations (orange circles). Each circle indicates one day of the JAS-season averaged over the WA-domain outlined in Fig. 1. The Panels show the following relationships: (a) the ratio between runoff and precipitation rate as function of precipitation rate, (b) the daily-change of soil moisture (within the six uppermost soil layers reaching up to a depth of 3.42m) as function of the runoff-precipitation ratio, (c) the difference in latent heat flux as function of the difference in total soil moisture content (again within the six uppermost soil layers reaching up to a depth of 3.42m) and (d) the difference in the precipitation rate as function of the difference of latent heat flux.

in past studies (Schär et al. (1999); Hohenegger et al. (2009)) or from a weaker change in latent heat flux due to a missing refilling of the soil moisture by precipitation. Fig. 10 c and d emphasize that the latter effect (weaker change in latent heat flux due to smaller changes in soil moisture) dominates the weaker response of precipitation to an increase in vegetation cover in the 5km-E simulations. While the response of precipitation to the change in latent heat flux is not substantially different in the 5km-E as compared to the 40km-P simulations (Fig. 10 d), Fig. 10 c indicates that much smaller changes in soil moisture yield
smaller changes in latent heat flux. We hypothesize that the higher soil moisture values in the explicitly resolved convection



simulations, comparable to the ones in the parameterized convection simulations, will enhance the land-atmosphere coupling and shift monsoonal precipitation further north.

## 3.3 The influence of runoff-controlled soil moisture

In the following, we test our hypothesis that the high amounts of runoff influence the precipitation response in the 5km-E
GS simulation via the control of the soil moisture. We eliminate the limiting influence of the runoff on soil moisture in both representations of convection. For this purpose, we perform a set of simulations where we prescribe the same soil moisture fields in the 40km-P and the 5km-E GS simulations and keep this soil moisture field constant for the whole simulation period (Sec.2.2.2). We refer to these simulations as GS-cSM ("Green Sahara with constant soil moisture").

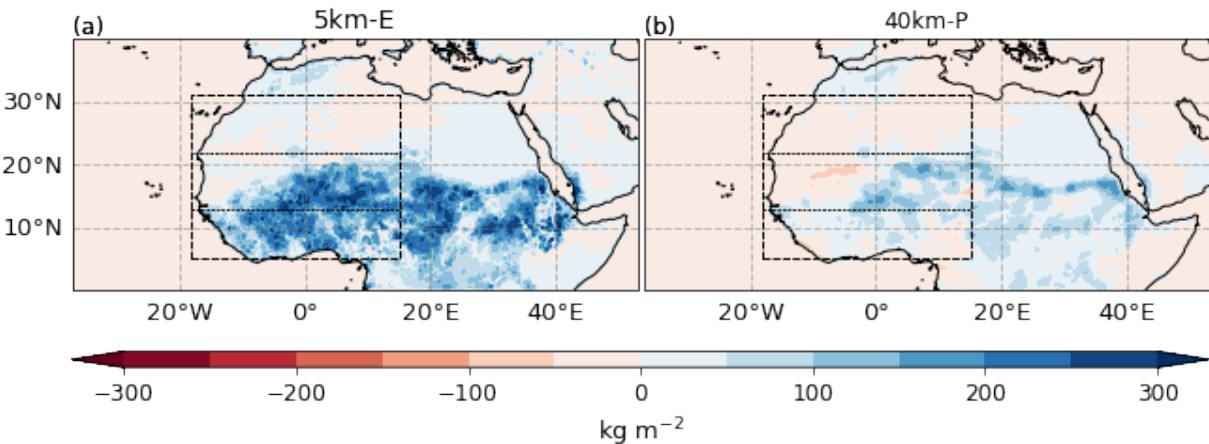

**Figure 11.** Difference of the JAS-mean soil moisture content in the six uppermost soil layers between the GS-cSM and the GS simulation for the 5km-E (a) and the 40km-P (b) simulations.

Figure 11 a shows that soil moisture noticeably increases in the 5km-E GS-cSM simulation while the difference between the
GS-cSM and the GS simulation in the 40km-P run is much smaller (Fig. 11 b). As a result, the latent heat flux (Figure 12a) strongly increases in the 5km-E GS-cSM simulation, especially over the Sahel region. In response to the strong increase in soil moisture and latent heat flux, especially over the Sahel and Sahara region, we find that the lower tropospheric humidity increases more strongly in the 5km-E GS-cSM simulations (not shown), supporting the triggering of convection. Further, the stability of the atmosphere especially over the Sahel decreases and conditions become more supportive in the 5km-E GS-cSM
as compared to the GS simulation (see Table A3 in the supplementary).

The increase in soil moisture and latent heat flux further reduces the lower-tropospheric temperature (Figure 12 b) over the Sahel in the 5km-E GS-cSM simulation and thereby further reduces the temperature gradient over the continent. As explained in Sec. 3.1.2, the temperature gradient strongly influences the AEJ strength and its location. The temperature reduction between the 5km-E GS-cSM and GS simulation is stronger than in the 40km-P simulations. As a result, the mean winds of the AEJ



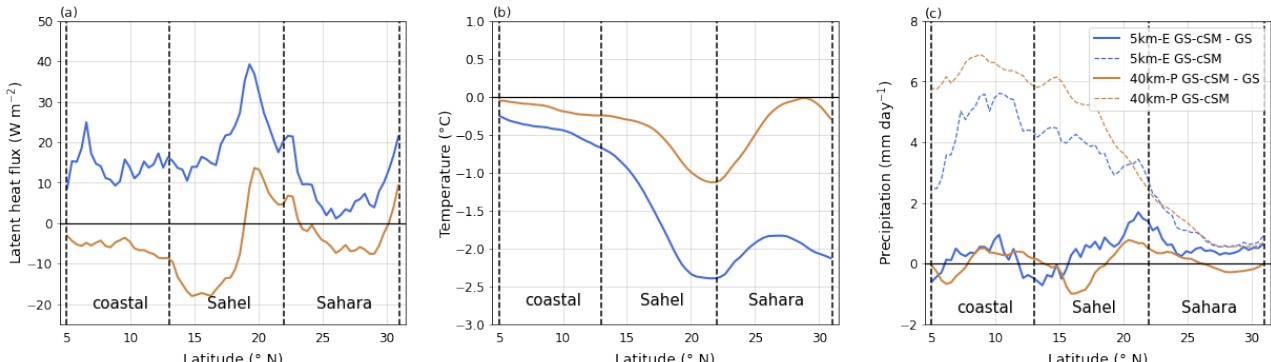

**Figure 12.** Difference of the JAS-mean meridional distributions of latent heat flux (a), 850hPa-temperature (b) and precipitation (c) between the GS-cSM and the GS simulations (solid lines) and for the GS-cSM simulation (dashed lines in Panel c) for the 5km-E (blue lines), and the 40km-P (orange lines), respectively.

weakens by $2\,\mathrm{m\,s^{-1}}$ in the 5km-E GS-cSM simulation (compared to the GS simulation) and by $0.5\,\mathrm{m\,s^{-1}}$ in the 40km-P GS-cSM simulations. In the end, the AEJ becomes weaker in the 5km-E GS-cSM ($6.6\,\mathrm{m\,s^{-1}}$) than in the 40km-P GS-cSM simulation ($7.9\,\mathrm{m\,s^{-1}}$). Moreover, the AEJ core shifts further north to 20.8°N in the 5km-E GS-cSM simulation and to 20.4°N in the 40km-P GS-cSM simulation. Hence, the AEJ is located at about the same location in both the 5km-E and 40km-P GS-cSM simulations. We argue that the location and strength of the AEJ are essential for the northward extent of monsoonal

precipitation in our simulations, consistent with the findings of Nicholson and Grist (2001), Grist and Nicholson (2001) and Cook (1999).

     Consequently, over the northern Sahel, precipitation (Fig. 12 c) shows a strong increase of up to about $1.7\,\mathrm{mm\,day^{-1}}$ in the 5km-E GS-cSM simulations, while the increase in the 40km-P GS-cSM simulation is noticeably smaller with about $0.8\,\mathrm{mm\,day^{-1}}$. As a result, monsoonal precipitation extends equally far north in the 5km-E GS-cSM and the 40km-P GS-

cSM simulation (Fig. 12 c, dashed lines). We, therefore, argue that the soil moisture north of about 17°N becomes an important factor for the northward propagation of monsoonal precipitation. This confirms our hypothesis that, in the 5km-E DS and GS simulations, the limited soil moisture hampers the northward extent of monsoonal precipitation in response to a change in vegetation cover.

     However, south of about 17°N precipitation remains much lower in the 5km-E simulations than in the 40km-P simulations.

This is likely because precipitation in this region is not moisture-controlled. Whether convection and precipitation can develop is determined by other processes, such as the presence of strong enough vertical lifting. Fig. 13 displays a much stronger vertical velocity in the 40km-P GS-cSM than in the 5km-E GS-cSM simulations. This is also true for the DS and the GS simulations, as well as for all horizontal resolutions. Therefore, we argue that the generally higher mean precipitation rates in the 40km-P simulations result from the convective parameterization scheme used in the ICON-NWP model framework.

The influence of precipitation via the thermodynamic and dynamic state of the atmosphere in the 5km-E GS-cSM compared to the GS simulation show the strong indirect effect of increased soil moisture on the atmospheric state. To quantify the



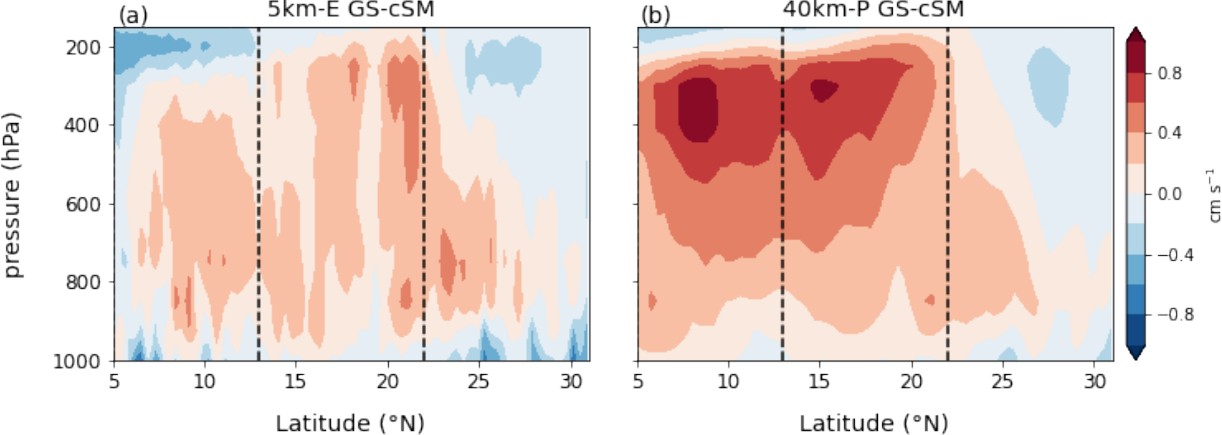

**Figure 13.** JAS-mean meridional cross-section of vertical wind speed for the 5km-E GS-cSM (a) and the 40km-P GS-cSM (b) simulation. The vertical dashed lines indicate the boarders of the coastal, Sahel and Sahara region outlined in Fig. 1.

contributions of the thermodynamic and the dynamic feedback, more simulations are necessary and are beyond the scope of this paper.

## 4 Summary and Conclusion

In this study, we have performed the first storm-resolving simulations (5km-E) of mid-Holocene North Africa with an idealized but reasonably increased vegetation cover. To investigate the land-atmosphere feedbacks we have compared these vegetated-Sahara simulations (GS) with simulations with mid-Holocene atmospheric conditions but with present-day vegetation cover (DS, see Jungandreas et al. (2021) for more details). In response to the higher vegetation cover, precipitation shifts from the coastal towards the Sahel and Sahara region in the GS simulation, and leads to a stronger northward extent of monsoonal precipitation of about 4-5° (regardless of the representations of convection). Our modelling results suggest, that both thermodynamic and dynamic feedbacks modulate precipitation over (mid-Holocene) North Africa. Which one dominates under which conditions needs to be further investigated.

We have identified the following feedback mechanisms:

1) As a result of the increased vegetation cover, soil moisture and interception water increases (also because precipitation increases), this yields higher latent heat fluxes over the Sahel and Sahara region. A higher latent heat flux increases the boundary-layer atmospheric moisture and generates more supportive thermodynamic conditions favourable for convection to develop.

2) The change in vegetation cover alters the dynamics of the monsoon circulation by affecting the temperature and moisture gradient over North Africa. The decrease of the temperature and moisture gradient result in a weakening and northward

shift of the AEJ. These changes in the AEJ lead to moister conditions over the Sahel in the whole troposphere, by both, enhanced upward transport (together with the TEJ) of boundary layer moisture and decreased, mid-level export of moisture from the African continent. These findings are consistent with previous results of Cook (1999), Grist and Nicholson (2001) and Nicholson and Grist (2001) who have associated a weaker and further northward located AEJ with more humid conditions over the Sahel.

When compared to simulations that use parameterized convection (40km-P) (as all global climate models, such as the ones used in the Paleo Modelling Intercomparison Project (PMIP), do up to now), we find important differences between the 40km-P and the 5km-E simulations. As shown in Jungandreas et al. (2021), the representation of convection strongly influences the hydrological cycle in our simulations. As a result of different precipitation characteristics (e.g. intensity and spatial distribution), the representation of convection influences the response of soil moisture to precipitation via the modulation
of runoff. We again emphasize that the land-atmosphere feedback is not only a result of how strong precipitation changes due to a certain change in latent heat flux or soil moisture. Also how the land surface (soil moisture and runoff) reacts to specific precipitation characteristics (drizzle or shower) needs to be taken into account in the feedback loop.

The consequently lower soil moisture values in the 5km-E simulation induce a weaker response of precipitation to a change in vegetation cover. To have a closer look at this difference in the response, we conducted simulations with the same constant
soil moisture field in both, the 5km-E and the 40km-P simulation. The elimination of the impact of runoff on soil moisture induces stronger changes in the 5km-E than in the 40km-P simulations, especially over the northern Sahel and Sahara region. This result confirms the impact of the land surface on precipitation over the Sahel and Sahara in our simulations. In the simulations with explicit convection, rainfall over the Sahel and Sahara tends to increase, while in parameterized convection simulations rainfall becomes smaller over the Sahel region and larger over the northern Sahel and the Sahara. Specifically,
the meridional difference between rainfall over the coastal region and the Sahara is smaller in the simulations with explicitly resolved convection.

This study highlights the importance to consider both pathways of the soil moisture-precipitation feedback: not only the precipitation response to changes in soil moisture conditions is important, but also the soil moisture response to specific precipitation characteristics plays a crucial role to maintain a strong positive feedback loop. The latter suggests that the repre-
sentation of the land surface in modeling studies, especially the soil hydrology (including runoff), is of major importance for an adequate representation of land-atmosphere interactions and other atmospheric processes. Describing realistic precipitation characteristics by simulating with explicit convection, specifically deep convection, can have far-reaching consequences for simulating land surface - atmosphere interaction and should be considered in future studies.

*Code availability.* http://hdl.handle.net/21.11116/0000-000A-E8FB-6

*Data availability.* Will be provided as soon as possible.



## Appendix A: Additional Figures and Tables

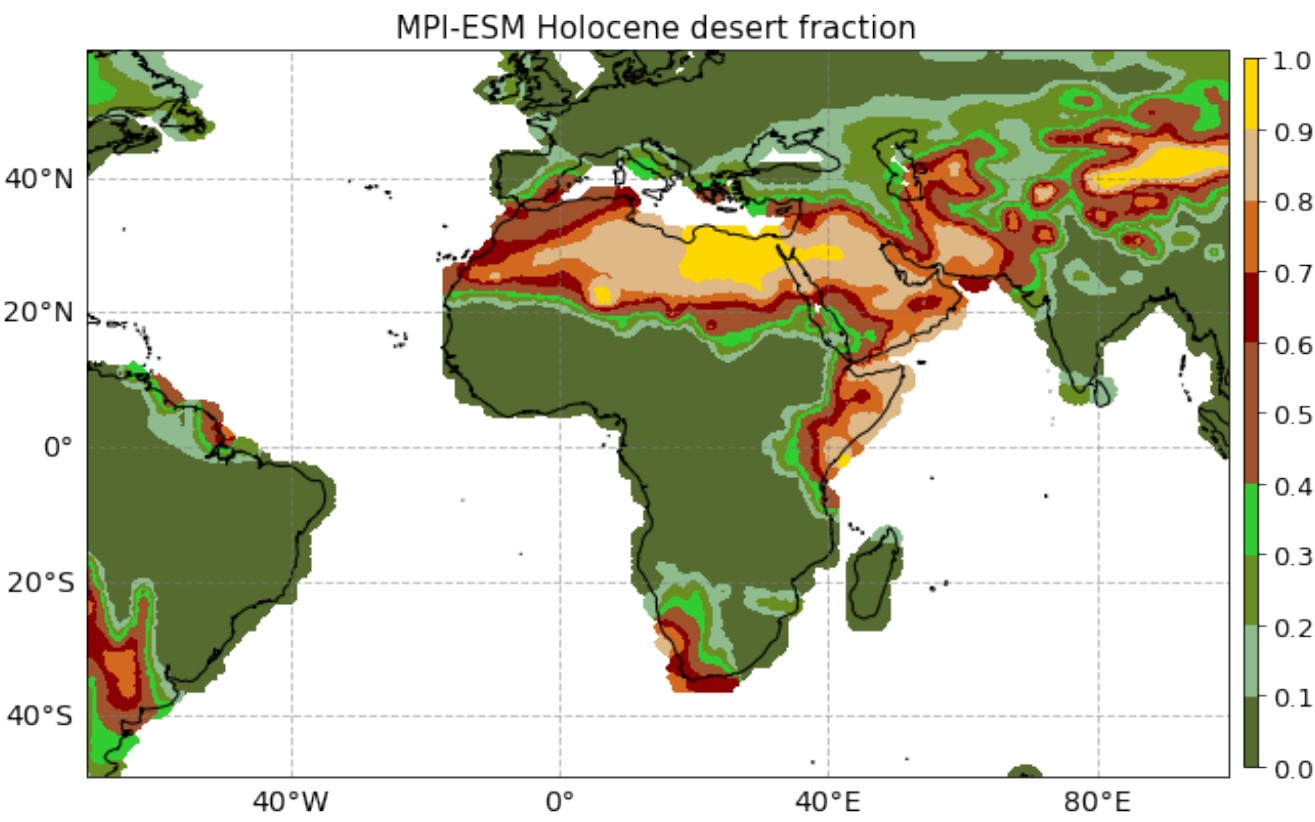

**Figure A1.** MPI-ESM desert fraction from the transient Holocene simulations. 0 - indicates no desert area within a grid box, 1 - the whole grid box is desert.





| Region | value (range) of desert fraction (MPI-ESM simulations) | present-day vegetation type | new vegetation type |
|---|---|---|---|
| North Africa 0°N - 37°N, 20°E - 50°W | <0.2 | 0,13,14,19 | 12 (closed to open shrublands) |
| | 0.2 - 0.6 | 19 | 13 (closed to open herbaceous vegetation) |
| | 0.6 - 0.9 | 19 | 14 (sparse vegetation) |
| | >0.9 | 0, 19 | 19 (bare areas) |
| | - | 1,2,3,4,5,6,11,12,15,17 | 4 (closed broadleaved evergreen forest) |
| | - | 8,10 | 12 (closed to open shrublands) |
| Europe and Eurasia north of 37°N | <0.1 | 1,2,3 | 9 (broadleaved and needleleaved forest) |
| | 0.1 - 0.4 | 1,2,3 | 12 (closed to open shrublands) |
| | 0.4 - 0.9 | 1,2,3 | 14 (sparse vegetation) |
| | >0.9 | all | 19 (bare areas) |
| South America west of 20°W, south of 20°N | <0.1 | all | 4 (closed broadleaved evergreen forest) |
| | 0.1 - 0.4 | all | 12 (closed to open shrublands) |
| | 0.4 - 0.9 | all | 14 (sparse vegetation) |
| | >0.9 | all | 19 (bare areas) |
| South Africa east of 20°W, south of 10°S | - | 3,10,12,13,15 | 4 (closed broadleaved evergreen forest) |
| South Africa east of 20°W, south of 20°N | - | 0,1,2,3 | 5 (closed broadleaved deciduous forest) |
| | - | 13,14,17,19 | 12 (closed to open shrublands) |
| India and Asia east of 60°E, south of 40°N | <0.1 | all | 4 (closed broadleaved evergreen forest) |
| | 0.1 - 0.4 | all | 12 (closed to open shrublands) |
| | 0.4 - 0.9 | all | 14 (sparse vegetation) |
| | >0.9 | all | 19 (bare areas) |
| Morocco from 10°W to 10°E, from 27°N to 37°N | - | 1,2,3,5,6,14 | 9 (mixed broadleaved and needleleaved forest) |
| North Arabia from 20°E to 60°E, from 20°N to 40°N | - | 0,1,2,3 | 9 (mixed broadleaved and needleleaved forest) |

**Table A1.** Conditions for the prescription of the new vegetation types for the "Green Sahara" simulations. We prescribe a new, idealized vegetation cover that reflects mid-Holocene conditions, depending on 1. the region, 2. the value range of the simulated desert fraction from the MPI-ESM Holocene simulations Dallmeyer et al. (2020), and 3. the present-day vegetation type default from IFS-reanalysis data.





| Modified variables | set to constant value |
|---|---|
| Plant cover maximum (PLCOV_MX) | mean |
| Leaf area index maximum (LAI_MX) | 85%-percentile for vegetation type 4 (closed broadleaved evergreen forest)<br>75%-percentile for all other vegetation types |
| Minimal stomata resistance (RSMIN) | mean |
| Longwave surface emissivity (EMIS_RAD) | mean |
| Root depth (ROOT_DP) | mean |
| Normalized difference vegetation index maximum (NDVI_MAX) | 85%-percentile for vegetation type 4 (closed broadleaved evergreen forest)<br>75%-percentile for all other vegetation types |
| Roughness length (Z0) | 85%-percentile for vegetation type 4 (closed broadleaved evergreen forest)<br>75%-percentile for all other vegetation types |
| Fraction of evergreen forest (FOR_E) | set to 0.88 for each grid box with the vegetation type 4 or 7; 0 for all other vegetation types |
| Fraction of deciduous forest (FOR_D) | set to 0.88 for each grid box with the vegetation type 5 or 6; 0 for all other vegetation types |
| Normalized difference vegetation index (NDVI) | 85%-percentile for vegetation type 4 (closed broadleaved evergreen forest)<br>75%-percentile for all other vegetation types |
| Normalized difference vegetation index - (monthly) proportion of actual value/maximum normalized differential vegetation index (NDVI_MRAT) | 85%-percentile for vegetation type 4 (closed broadleaved evergreen forest)<br>75%-percentile for all other vegetation types |
| Surface albedo (ALB) | mean |
| Near infrared albedo (ALNID) | mean |
| UV Albedo (ALUVD) | mean |

**Table A2.** Values prescribed for each variable and each vegetation type in the GS simulations. The values were calculated based on all grid points where a vegetation type is dominant.





|  |  | 40 km-P DS | 40 km-P GS | 40 km-P GS-cSM | 5 km-E DS | 5 km-E GS | 5 km-E GS-cSM |
|---|---|---|---|---|---|---|---|
| coastal | CAPE (J kg$^{-1}$) | 737.2 | 561.1 | 503.9 | 706.7 | 505.6 | 718.8 |
|  | CIN (J kg$^{-1}$) | -29.4 | -25.7 | -22.6 | -37.5 | -34.1 | -18.6 |
|  | LFC (m) | 724 | 652 | 683 | 1131 | 1236 | 918 |
|  | cloud cover (%) | 87 | 89 | 90 | 77 | 79 | 80 |
| sahel | CAPE (J kg$^{-1}$) | 382.1 | 500.7 | 474.1 | 159.8 | 220.6 | 498.8 |
|  | CIN (J kg$^{-1}$) | -232.8 | -232.8 | -84.7 | -262.0 | -211.5 | -94.1 |
|  | LFC (m) | 2280 | 1482 | 1125 | 2617 | 1940 | 1208 |
|  | cloud cover (%) | 44 | 68 | 73 | 36 | 58 | 67 |
| sahara | CAPE (J kg$^{-1}$) | 0.0 | 12.1 | 25.5 | 0.0 | 2.1 | 6.0 |
|  | CIN (J kg$^{-1}$) | 0.0 | -106.1 | -102.9 | 0.0 | -28.2 | -44.4 |
|  | LFC (m) | 3687 | 2804 | 2693 | 3321 | 2973 | 2506 |
|  | cloud cover (%) | 17 | 34 | 36 | 20 | 36 | 43 |

**Table A3.** 12 UTC JAS-mean values of CAPE, CIN and JAS-mean values of the level of free convection (LFC) and total cloud cover for the coastal, Sahel and Sahara region (Fig. 1) and for the 40 km-P DS, GS and GS-cSM simulation and for the 5 km-E DS, GS and GS-cSM simulation.



## Appendix B: Analysis of 10km-simulations

We analyze the effect of the horizontal resolution on our findings and present the results for the 10km parameterized (10km-P) and explicitly resolved convection (10km-E) simulations in the following. We show that the main mechanisms described for the 5km-E and the 40km-P simulations are also valid for the 10 km-E, and 10 km-P simulations. The results between simulations with the same representation of convection are more similar to each other than simulations with the same horizontal resolution. However, the results show that differences between the 10km-E and the 10km-P simulations are not as strong as between the 5km-E and the 40km-P simulation. We do find the characteristic precipitation-runoff-soil moisture mechanism that limits the soil moisture-precipitation feedback in the 10km-E DS and GS simulations but the difference to the 10km-P simulations coupling is not as strong. We argue that this is because, in the 10km-P simulation, the grid spacing already allows partly for explicit calculations of precipitation as compared to the 40km-P simulations. Therefore, precipitation intensity and runoff are more similar in the 10km-E and 10km-P simulations than in the 5km-E and the 40km-P simulations.

### B1 Changes in vegetation, latent heat flux and temperature

Fig. B1 shows dominantly similar responses of the surface variables in the 10km-P and 10km-E simulations as in the 40km-P and 5km-E simulations, respectively. The meridional vegetation gradient (see Fig.4a) is identical to the one in the 5km-E and the 40km-P simulation because it is prescribed in the external parameters. The largest differences between the GS and DS simulations are simulated over the Sahel region, similarly to the results in the main paper. The increased vegetation cover yields higher water availability and, hence, leads to a decrease in sensible heat flux (Fig. B1a,b) and an increase in latent heat flux (Fig. B1c,d) over the Sahel. Over the Sahara region, the increase in sensible heat flux dominates the increase in the turbulent heat fluxes, because water becomes too limited to support stronger latent heat flux. This increase is approximately equally strong in the 10km-E and the 10km-P simulations. We notice that differences in the turbulent heat fluxes between the 10km-E and the 10km-P simulations are less pronounced than between the 5km-E and the 40km-P simulations. The main difference is that the maximum increase in latent heat flux in the 10km-P is located further north (at about 19°N) as compared to the 10km-E simulations (at about 17°N), with implications for temperature and moisture gradient over the continent.

The temperature over the North African continent decreases and the location of the maximum temperature shifts northward (about 2°) in both representations of convection. Changes in the meridional temperature distribution between the GS and DS simulations are noticeably stronger in the 10km-P than in the 10km-E simulations. The 10km-P simulations display a stronger maximum decrease in temperature (about 3°C) located at about 19°N (consistent to the location of the maximum increase in latent heat flux). In contrast, the maximum temperature decrease in the 10km-E simulations is only 2°C and located further south at about 17°N. As already shown in the main paper, this can have far-reaching consequences for the monsoon dynamics linked to the AEJ.







**Figure B1.** JAS-mean meridional distribution of sensible heat flux (a,b), latent heat flux (c,d), and 850hPa-Temperature (e,f) for the Desert Sahara simulations (blue dotted lines), the Green Sahara simulations (blue dashed lines), and the difference between Green Sahara and Desert Sahara simulation (blue solid lines) for the 10km-P (a,c,e) and the 10km-E (b,d,f) simulation, respectively. The vertical dashed lines indicate the boarders of the coastal, the Sahel, and the Sahara region outlined in Fig.1.

## B2 Changes in atmospheric dynamics

The low-level southwesterly monsoon intensifies as a response to the increased vertical motion over the Sahel (due to increased energy at the surface and by the increase of vertical motion between the jet axes). With the strengthened monsoon flow, more
420 moisture is transported from the coastal region deep into the Sahel and Sahara region, contributing to the decrease in relative humidity (Fig. B4). Moreover, the ITF shifts further north and thereby the associated region of dynamical uplifting caused by





the convergence of the southwesterly monsoon winds and the northeasterly Harmattan winds shift further north. Similar to the 40km-P simulations, the southwesterly monsoon flow is stronger and the monsoon layer is deeper in the 10km-P compared to the 10km-E simulations. Therefore, the ITF is located further north and more cool, moist air is transported into the African continent to support the generation of convection and precipitation.

In response to the changes in the temperature gradient over North Africa, the AEJ (Fig. B2) strongly weakens in the GS simulations and shifts 3-4° northward in both representations of convection. With the shift in the AEJ, the region of upward motion between the axes of the TEJ and the AEJ broaden/shift northward. Consequently, in both representations of convection, the vertical upward motion (Fig. B3) decreases over the coastal and southern Sahel region and intensifies over the rest of the Sahel region. Furthermore, the descending motion over the Sahara decreases in the GS compared to the DS simulations. The upward vertical motion is generally stronger in the 10km-P than in the 10km-E simulations (Fig. B3a-d). We can also find the generally stronger vertical upward motion in the 40km-P as compared to the 5km-E simulation. We therefore suggest that the generally stronger vertical upward motion is a result of the convective parameterization scheme in our simulations. Further investigations of this feature would be necessary to confirm and explain this assumption.

### B2.1 Changes in atmospheric thermodynamics

Changes in relative humidity in the 10km-P and the 10km-E simulation are consistent with the changes in the 40km-P and the 5km-E simulation, respectively. Relative humidity reveals a strong increase over the Sahel and Sahara region, which is consistent with the changes in latent heat flux (Fig. B1 c,d) and in the wind field (Fig. B2). Again, it is likely that stronger convection in the GS simulations positively feeds back on the atmospheric humidity.

Generally, the 10km-P simulations show higher relative humidity values throughout the troposphere and overall latitudes (not shown). The increase in relative humidity over the Sahel-Sahara region between the GS and the DS simulation is more pronounced in the 10km-P GS than in the 10km-E GS simulation, while the decrease in relative humidity over the coastal region is stronger in the 10km-E GS simulation.

Table B1 reveals that the stability of the atmosphere becomes more supportive for convection and precipitation in the GS compared to the DS simulations in both representations of convection. Over the coastal region, conditions for convection and precipitation are very supportive, with high CAPE and cloud cover and low (weak negative) CIN and LFC values. We find that CAPE over the Sahel increases more strongly in the 10km-E GS than in the 10km-P GS simulation. Additionally, CIN values are less negative. However, the LFC is lower and cloud cover higher in the 10km-P than in the 10km-E simulations.

Conditions for convection and precipitation are overall more supportive in the 10km-P simulations. However, differences in CAPE between the 10km-P GS and the 10km-E GS simulation are smaller or even reversed as compared to the results between the 40km-P GS and the 5km-E GS simulation.







**Figure B2.** The same as Fig. 5 but for the 10km-P DS (a) and GS (c) simulation and the 10 km-E DS (b) and GS (d) simulation, respectively.

## B3 Changes in precipitation

In agreement with the results of the main Paper (Sec. 3.1.4), precipitation (Fig. B5) in the GS simulations decrease over the coastal region. This decrease is consistent with the decrease in latent heat flux (Fig. B1 c,d), in relative humidity (Fig. B4) and in vertical upward motion (Fig. B3). The strongest increase in precipitation occurs over the Sahel region (Fig. B5 and B6),





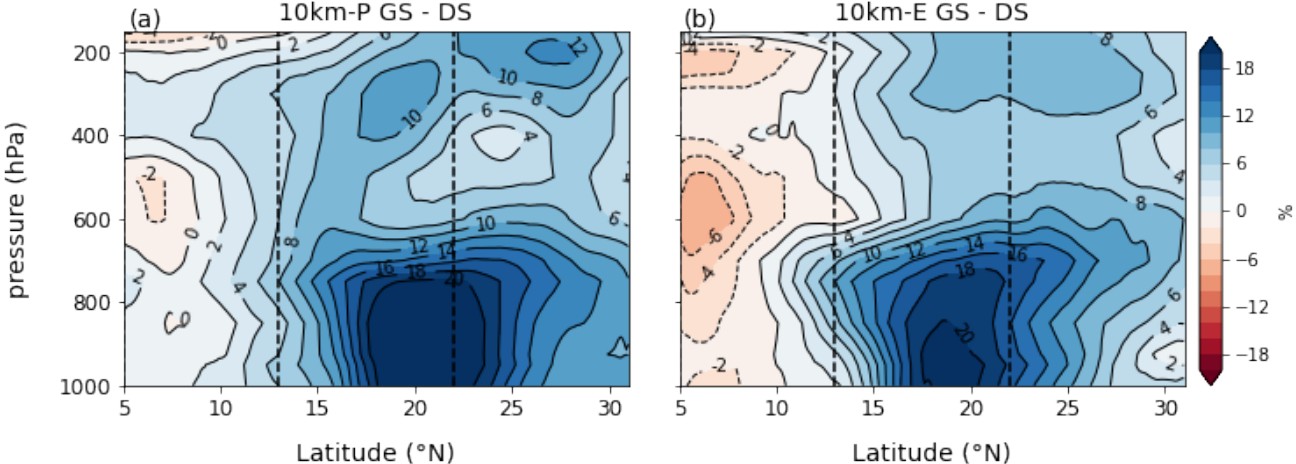

**Figure B3.** JAS-mean meridional vertical crosssection of vertical velocity for the 10km-P DS (a) and GS (c) simulation, the 10km-E DS (b) and GS (d) simulation and the difference between the GS and DS simulation for the 10km-P (e) and the 10km-E (f) simulations. Positive values indicate upward motion. The average is calculated over the WA-Domain outlined in Fig. 1. Note the different color-scale in panel e and f.

|  |  | 10 km-P DS | 10 km-P GS | 10 km-E DS | 10 km-E GS |
|---|---|---|---|---|---|
| coastal | CAPE (J kg$^{-1}$) | 641.5 | 505.0 | 818.5 | 567.9 |
|  | CIN (J kg$^{-1}$) | -30.7 | -22.8 | -31.0 | -31.2 |
|  | LFC (m) | 738 | 654 | 945 | 1000 |
|  | cloud cover (%) | 88 | 91 | 79 | 78 |
| sahel | CAPE (J kg$^{-1}$) | 309.6 | 390.6 | 162.6 | 300.2 |
|  | CIN (J kg$^{-1}$) | -257.6 | -291.2 | -128.1 | -199.2 |
|  | LFC (m) | 2253 | 1365 | 2648 | 1902 |
|  | cloud cover (%) | 48 | 74 | 34 | 57 |
| sahara | CAPE (J kg$^{-1}$) | 0.8 | 13.01 | 0.0 | 0.7 |
|  | CIN (J kg$^{-1}$) | -43.7 | -83.3 | 0.0 | -4.3 |
|  | LFC (m) | 3508 | 2692 | 3203 | 2907 |
|  | cloud cover (%) | 18 | 37 | 17 | 33 |

**Table B1.** The same as Table 2 but for the 10km-P and 10km-E simulations.

coinciding with the region of the strongest increase in latent heat flux, thermodynamic and dynamic conditions. The enhanced conditions for convection and precipitation over the Sahel and Sahara region are consistent with the weakening and northward





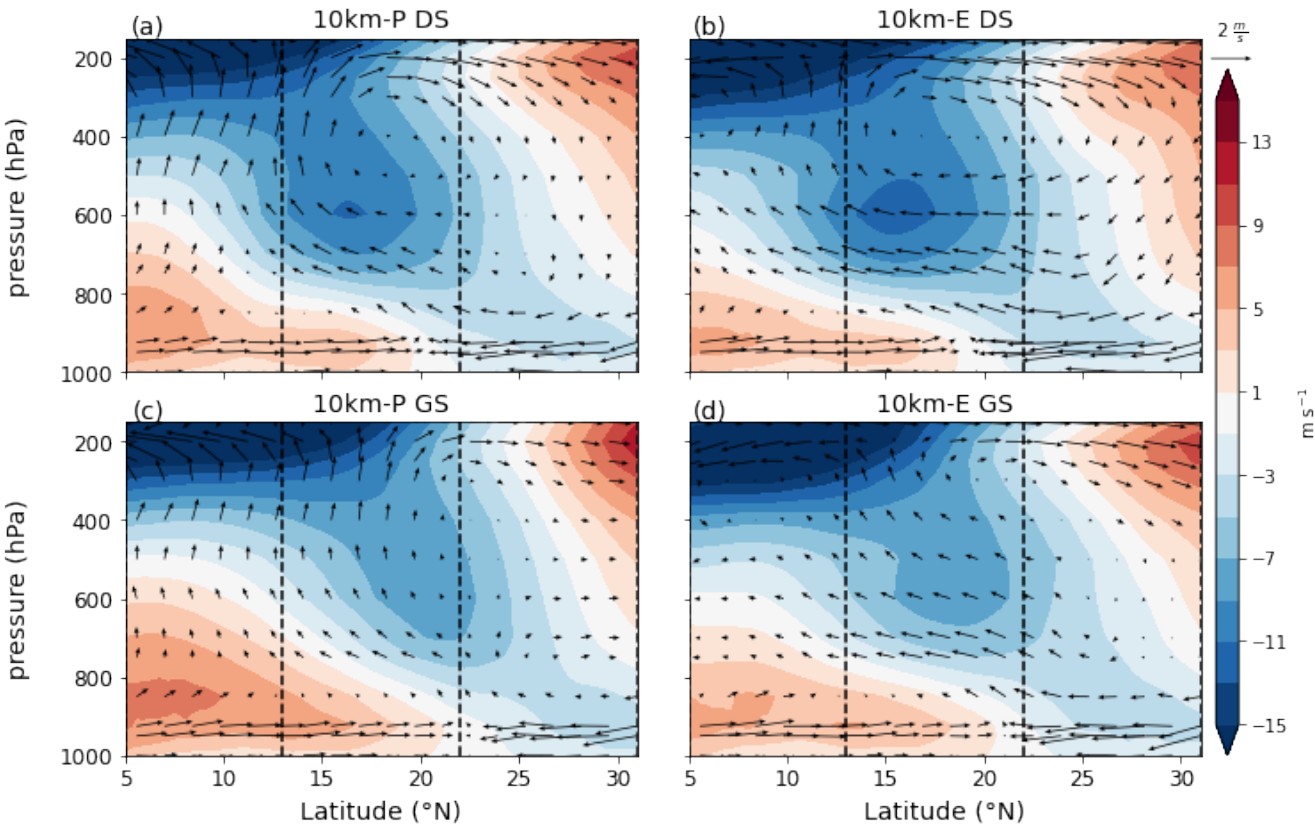

**Figure B4.** JAS-mean meridional vertical crosssection of relative humidity for the difference between the GS and DS simulation for the 10km-P (a) and the 10km-E (b) simulations. The average is calculated over the WA-Domain, and the vertical dashed lines indicate the boarders of the coastal, the Sahel, and the Sahara region outlined in Fig. 1).

shift of the AEJ and agree with results of previous studies (Grist and Nicholson, 2001; Nicholson and Grist, 2001). Similar to the precipitation response in the main paper, the 10km-P simulations simulate a stronger increase of precipitation over to Sahel

and Sahara region and a weaker decrease over the coastal region as compared to the 10km-E simulations.

Consistent with the findings in the main paper, both representations of convection simulate a positive land-atmosphere coupling in response to the increase in vegetation cover over the Sahel and Sahara region, while a negative feedback evolves over the coastal region. This land-atmosphere coupling is stronger in the 10km-P simulations compared to the 10km-E simulations. However, differences tend to be smaller (e.g. in latent heat flux and thermodynamics).

**B4    The soil moisture-runoff-precipitation mechanism**

We can find a more pronounced response of the atmosphere to changes in the vegetation cover in the 10km-P than in the 10km-E simulations. As already examined in the main paper, this weaker land-atmosphere coupling is strongly determined



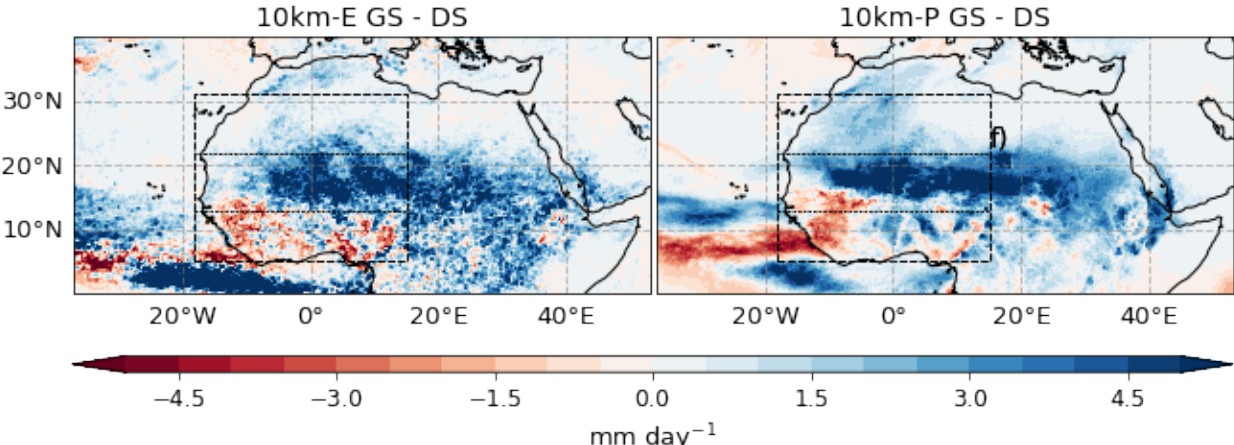

**Figure B5.** The same as Fig. 8 but for the 10km-P (a) and 10 km-E (b) simulations.

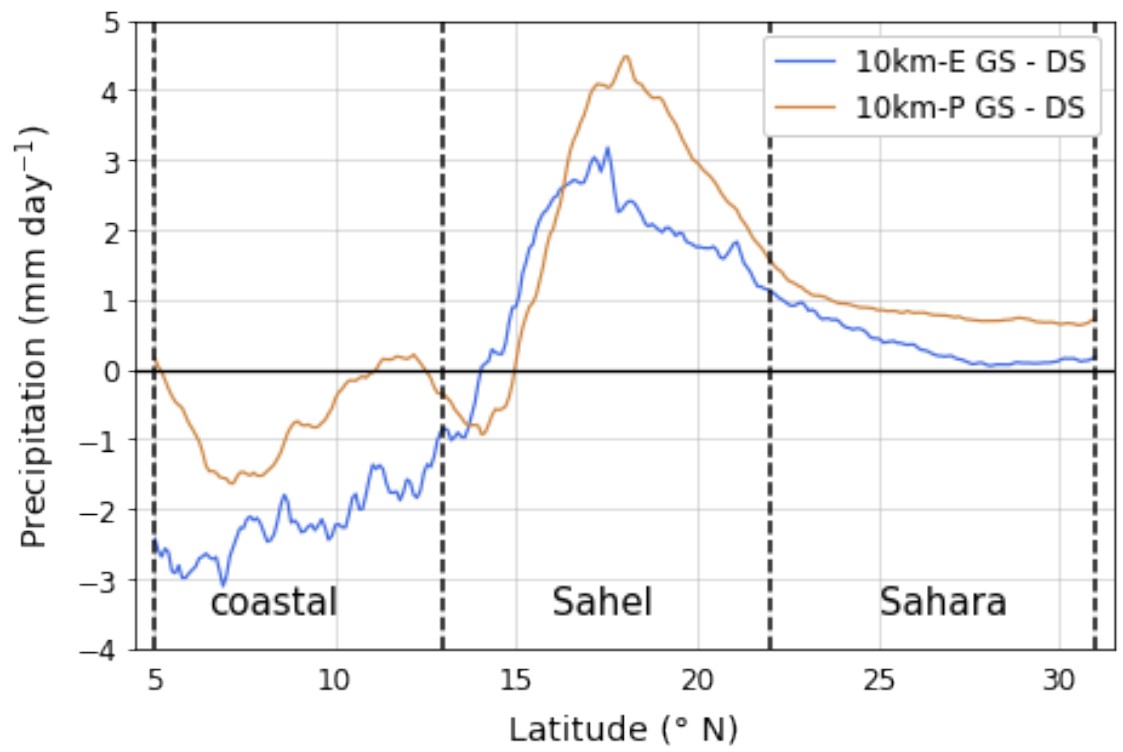

**Figure B6.** The same as Fig. 9 but for the 10km-P (orange line) and 10 km-E (blue lines) simulations.



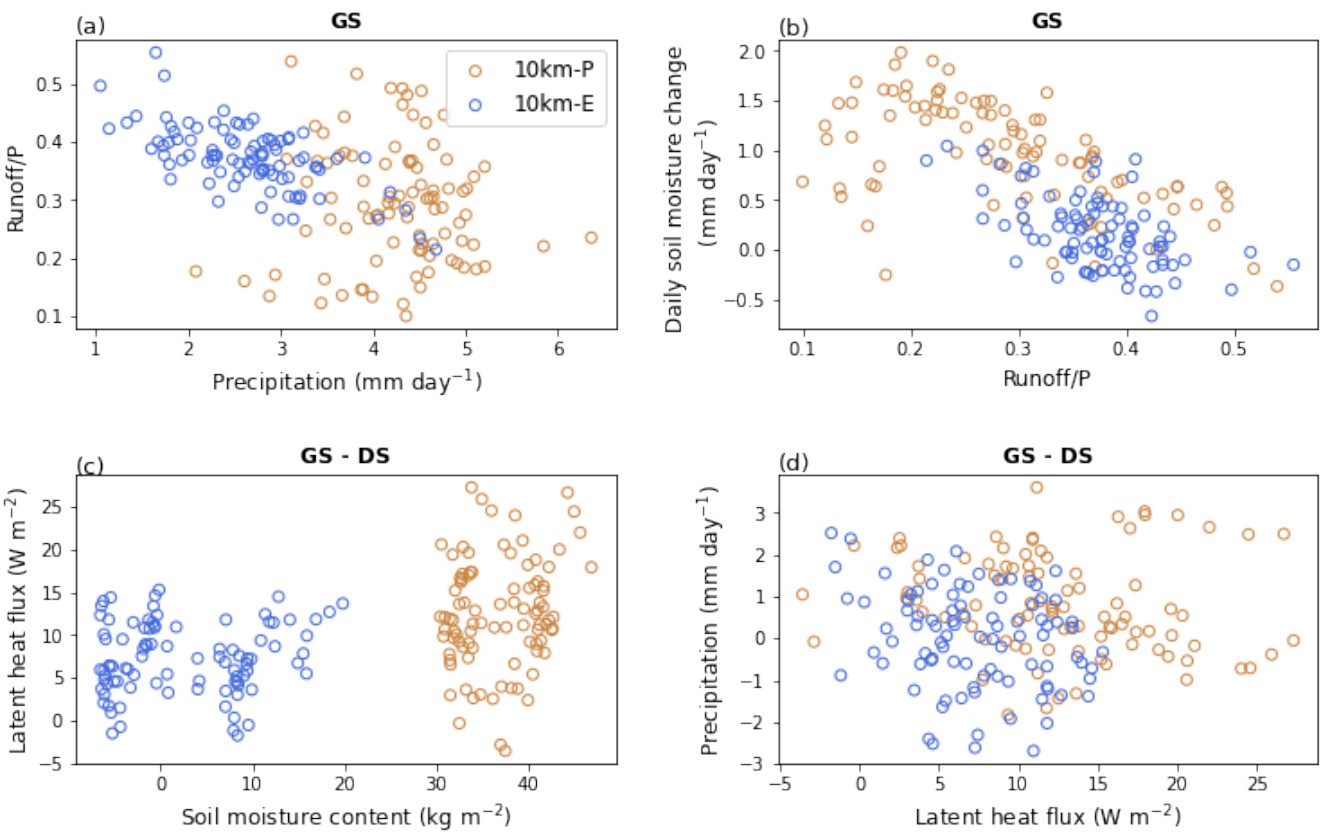

**Figure B7.** The same as Fig. 10 but for the 10km-P (orange circles) and 10 km-E (blue circles) simulations.

by the soil moisture-runoff-precipitation mechanism that dampens the potential precipitation response to changes in the land surface in explicitly resolved convection simulations.

Fig. B7 a displays higher amounts of runoff relative to the amount of precipitation in the 10km-E than in the 10km-P simulations. In both, the 10km-E DS (not shown) and GS simulation 36% of precipitation leave the system as runoff. In comparison, in the 10km-P DS (not shown) and GS simulation only 23% and 29% of precipitation are lost into runoff.

Due to the overall higher loss of precipitation water into runoff, the daily change in soil moisture content (in the uppermost six soil layers up to a depth of 3.42m) is smaller in the 10km-E than in the 10km-P simulations (Fig. B7 b). Hence, soil

moisture is generally noticeably lower in the 10km-E as compared to the 10km-P simulations (especially over the coastal and Sahel region) (not shown). Moreover, the changes in soil moisture between the GS and DS simulation are much smaller in the 10km-E than in the 10km-P simulations, resulting in smaller mean evapotranspiration rates (Fig. B7 c). This feature is identical to the results of the main paper. However, Fig. B7 d indicates that the reason for the weaker precipitation response in the 10km-E simulations is not because the coupling of precipitation to a given change in evapotranspiration is weaker. It is rather the

dampened evapotranspiration due to a missing refilling of soil moisture that leads to smaller precipitation rates. Additionally,





we argue that the weaker dynamic feedback contributes to smaller precipitation rates in the 10km-E simulations as compared to the 10km-P simulations. Based on these results, we argue that the precipitation-runoff-soil moisture mechanism also exists in the 10km-E simulations.

### B5   The influence of runoff-controlled soil moisture

We now analyze the differences between the 10km GS and GS-cSM simulations to investigate to which extent the runoff hampers the potential precipitation response to changing surface conditions.

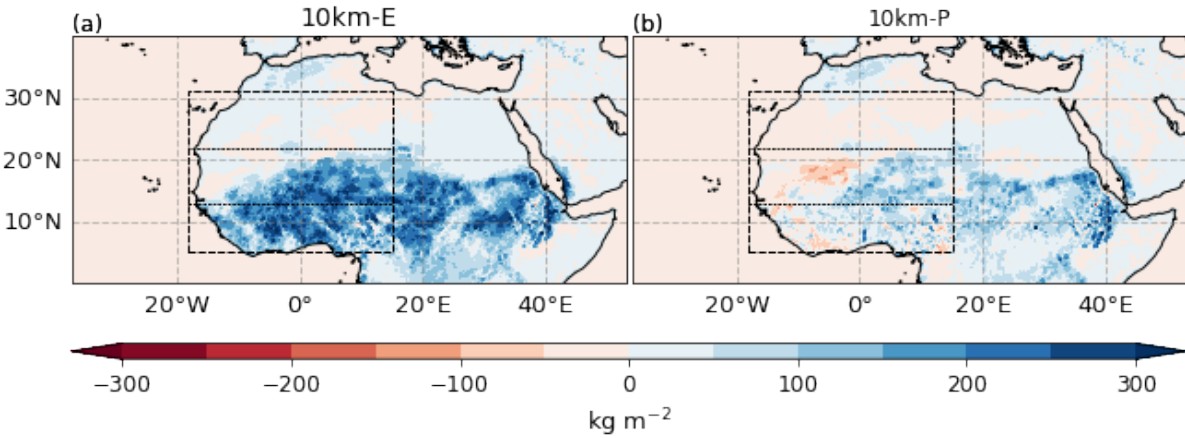

**Figure B8.** The same as Fig. 11 but for the 10km-E (a) and the 10km-P (b) simulation, respectively.

Similar to the results in the main paper, soil moisture content (Fig.B8) shows a strong increase over the whole WA-Domain in the 10km-E GS-cSM simulation (as compared to the GS simulation), while changes between the 10km-P GS-cSM and GS simulation are much smaller (or even show a decrease in soil moisture).

The resulting increase in latent heat flux in the 10km-E GS-cSM simulation is noticeably higher, with a maximum increase of $40 \, \mathrm{W} \, \mathrm{m}^{-2}$ over the northern Sahel, as compared to the 10km-P GS-cSM simulation (Fig. B9 a). This increase in latent heat flux modulates the increase in relative humidity (not shown) and promotes the increase in atmospheric thermodynamic conditions (Tabel B2) for convection and precipitation in the 10km-E GS-cSM simulations. Changes in CAPE, CIN, LFC height, and cloud cover between the GS-cSM and the GS simulations are stronger in the 10km-E than in the 10km-P simulations and further

support the increase in precipitation in the 10km-E GS-cSM simulation. In the 10km-P GS-cSM simulation, CAPE decreases slightly over the Sahel region, but CIN values, LFC height and cloud cover support the still higher precipitation rates in the 10km-P GS-cSM compared to the 10km-E GS-cSM simulations. Differences in the thermodynamics of the atmosphere between the 10km-P and the 10km-E simulations are weaker than between the 40km-P and the 5km-E simulations. Nevertheless, they reflect a noticeable influence of soil moisture on the atmosphere.





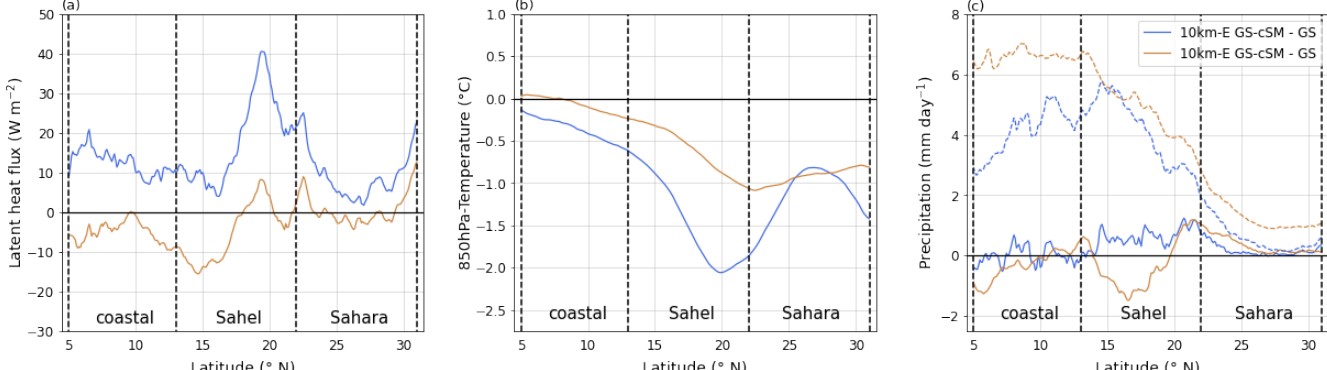

**Figure B9.** The same as Fig. 12 but the difference is calculated between the GS-cSM and the GS simulation for the 10km-P (a) and 10 km-E (b) simulations.

| | | 10 km-P GS | 10 km-P GS-cSM | 10 km-E GS | 10 km-E GS-cSM |
|---|---|---|---|---|---|
| coastal | CAPE (J kg$^{-1}$) | 505.0 | 423.8 | 567.9 | 792.4 |
| | CIN (J kg$^{-1}$) | -22.8 | -24.1 | -31.2 | -17.3 |
| | LFC (m) | 654 | 748 | 1000 | 781 |
| | cloud cover (%) | 91 | 90 | 78 | 79 |
| sahel | CAPE (J kg$^{-1}$) | 390.6 | 349.7 | 300.2 | 721.3 |
| | CIN (J kg$^{-1}$) | -128.1 | -92.27 | -199.23 | -84.72 |
| | LFC (m) | 1365 | 1076 | 1902 | 1269 |
| | cloud cover (%) | 74 | 76 | 57 | 66 |
| sahara | CAPE (J kg$^{-1}$) | 13.0 | 16.0 | 0.74 | 5.49 |
| | CIN (J kg$^{-1}$) | -83.3 | -119.52 | -4.26 | -77.35 |
| | LFC (m) | 2692 | 2448 | 2907 | 2730 |
| | cloud cover (%) | 37 | 41 | 33 | 38 |

**Table B2.** The same as Table A3 but for the 10km-P and 10km-E simulations.

As described in the main paper, the change in soil moisture and latent heat flux affect the temperature over North Africa (Fig. B9 b). The decrease in temperature is stronger in the 10km-E GS-cSM than in the 10km-P GS-cSM simulations, with the strongest decrease over the northern Sahel region. This much stronger decrease in temperature over the Sahel than over the coastal region modulates the temperature gradient over the continent and influences the atmospheric monsoon dynamics. As a result of the weakened temperature gradient south of about 20°N, the AEJ (not shown) again shifts almost 2° further

north in the 10km-E GS-cSM simulation, while in the 10km-P GS-cSM simulation it remains at about the same latitude as in





the GS simulation. The AEJ strength (not shown) does not change substantially between the GS-cSM and the GS simulation in both the 10km-E (about $8.3\,\mathrm{m\,s^{-1}}$ and $8.2\,\mathrm{m\,s^{-1}}$, respectively) and 10km-P simulations (about $7.9\,\mathrm{m\,s^{-1}}$ and $8.3\,\mathrm{m\,s^{-1}}$, respectively).

In the 10km-E GS-cSM simulation, the northward shift of the AEJ core leads to a weaker vertical upward motion in the lower and middle troposphere between about 18°-23°N (Fig. B10 c). However, air masses are still lifted in this region (Fig. B10 a) and the increasing vertical upward motion in the upper troposphere (Fig. B10 c, above 500hPa) indicates that air masses can be lifted further up. Hence, air masses can reach the freezing level more easily, which makes the generation of precipitation more effective. This is consistent with the maximum increase in precipitation (Fig. B9 c) in this region. Over the Sahara region vertical upward motion increases in the lower atmosphere layers and the descending motion in the upper troposphere decreases.

However, only in the southern Sahara, the availability of enough moisture allows for the formation of some precipitation.

In the 10km-P GS-cSM simulation, vertical upward motion increases noticeably from 20°-31°N, especially in the upper troposphere and strongly decreases in the middle and upper troposphere between 13-20°N (Fig. B10 d). This increase in vertical upward motion over the northern Sahel and Sahara region supports the strong increase in precipitation between 20°-25°N (Fig. B9 c). Generally, vertical upward motion remains substantially stronger in the 10km-P GS-cSM simulations over the

whole WA-Domain (Fig. B10 a and b) as already found in the DS and GS simulations (Fig. B3) and in the main Paper (Fig. 13).

Precipitation (Fig. B9 c) increases especially over the Sahel region in the 10km-E GS-cSM simulation, while in the 10km-P GS-cSM simulation, precipitation displays a strong decrease between 14°-20°N, consistent with the strong decrease in vertical upward motion (Fig. B10 d). However, over the northern Sahel (north of 20°N) and the southern Sahara region (up to about 25°N), precipitation shows an equally strong or stronger increase in precipitation in the 10km-P as compared to the 10km-

E simulations, despite only very weak changes in soil moisture and latent heat flux. Hence, in contrast to the results in the main paper, monsoonal rainfall in the GS-cSM simulations extend slightly further north in the 10km-P than in the 10km-E simulations.

In summary, the 10km-P and 10km-E simulations also show a positive land-atmosphere feedback over the Sahel-Sahara region, and a negative feedback over the coastal region. The differences between the 10km-E and 10km-P simulations are

slightly weaker than between the 5km-E and the 40km-P simulations but the principle mechanisms remain the same. The comparison between the GS and the GS-cSM simulations reveal a considerable influence of the runoff (i.e of soil moisture) on the atmospheric state, hence on monsoonal precipitation in our simulations.





**Figure B10.** The same as Fig. 6 but for the 10km-P GS-cSM (a) and the 10km-E GS-cSM (b) simulations. Additionally, the difference between the GS-cSM and the GS simulations is shown for the 10km-P (c) and 10km-E (d) simulations, respectively.





*Author contributions.* LJ, CH and MC together designed the research project and the experiments. LJ performed the simulations and analysis. CH and MC gave input, ideas and feedback to the analysis of the simulations. LJ prepared the manuscribt with contributions from all co-
authors.

*Competing interests.* The authors declare that they have no conflict of interest.

*Acknowledgements.* Will be written after internal review.



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
