# Peer review of "How does the explicit treatment of convection alter the precipitation-soil hydrology interaction in the mid-Holocene African humid period?"

_EGUsphere, 2022_

## Author Comment (AC1)

**Reply to Comments of Referee 1** to the Manuscript of Jungandreas et al. "How does the explicit treatment of convection alter the precipitation-soil hydrology interaction in the Holocene African humid period?"

Thank you for this very useful review!

**Main comments:**

1.) Throughout the text the authors refer to the period as "mid-Holocene" but in the title it says "Holocene" which of course is not wrong, but maybe a bit inconsistent.

*To avoid misunderstanding, we will change the title to "… mid-Holocene African humid period".*

2.) The manuscript starts with the term "storm-resolving" simulations in the second line of the abstract, which means resolved deep convection on the kilometer scale. However, the authors refer to it simply as "explicitly resolved convection" throughout the text, before calling it "deep-convection" in the last sentence of the conclusion. The terminology could be clarified in the beginning.

*We will clarify the terminology. As it has now become common, we will use the term "storm-resolving" throughout the manuscript to refer to our simulations. We will also define what we mean by storm-resolving at the beginning of the manuscript to avoid confusion.*

3.) The authors refer to the simulations with present day vegetation cover as "Desert Sahara". The word "Sahara" is derived from the Arabic word for desert, making the naming "Desert Deserts". A more applicable nomenclature without changing the abbreviation "DS" could be "Dry Sahara".

*This is a valid point and a good suggestion. We will use the term "Dry Sahara".*

4.) In line 102 the authors note: "We select two years after this 15-year soil moisture-spinup phase and start our nesting experiments for the boreal summer monsoon season." I would like to know why these two years were chosen and which of them was/were used in the analysis?

*We selected the two years based on the same procedure as in Jungandreas et al. (2021). Based on the 30-year monthly mean timeseries of the 40-km simulation, JAS-mean values and the latitudinal extent of the monsoon for every year over the WA-domain, we selected a relatively strong and a relatively weak monsoon year (in terms of precipitation amount and northward extent).*
*In the analysis we used the strong monsoon year.*
*We will clarify this in the manuscript.*

5.) I would be interested to know whether the precipitation in the GS and GS-cSM simulations is sufficient to sustain the prescribed vegetation cover.

*For a model-consistent answer, we would need to implement a vegetation model into the regional climate model, which is beyond the scope of this study.*

*However, a quick first guess could help. Vegetation needs roughly 200mm/year (Joussome et al, 1999) for a transition from desert-like vegetation to steppe.*

*If we assume that most, if not all, monsoon precipitation falls during the months from June to September, then a value of 200 mm/year corresponds to some 1.7 mm/d in our simulation. In Figure 12c, we see that the value of 1.7 mm/d is crossed at around 24°N in the GS-cSM simulation and a bit further south in the GS simulation. This is roughly consistent with the prescribed vegetation cover, keeping in mind that north of 24°N, the Libyan Sand Sea exists in the eastern Sahara, prescribed as bare area in the model. Hence in principle the precipitation would be sufficient to maintain the vegetation cover. Wew ill add this in the revised version.*

6.) In Figure 10 a) there seems to be a "separation" of 40 km-P data-points at about 10 % runoff and between 2 and 4 mm/day precipitation. This "separation" is also visible in Fig. 10 b) at about 10 % runoff with a wide range of corresponding soil moisture changes. I would be interested in an explanation for these results. Are these data- points related to a specific time during the JAS season (and thereby maybe also a region)? This is particularly interesting to me because, if these data-points are neglected, the precipitation (mm/day) to runoff (%) relationship appears to be similar for the 40 km-P and 5 km-E simulations in the overlapping range of 2.5 to 4.5 mm/day, i.e. all data-points roughly follow a linear trend.

*We find that these circles indeed correspond to the first 20 days of the analyzed period. During this period, soil moisture is still low compared to later in the monsoon season. Therefore soil moisture storage is not filled. Much of the widespread light precipitation (drizzle) can be taken up by soil moisture. Later in the season when soil moisture is higher, much more of precipitation (also in form of drizzle) goes into runoff.*
*However, we could not identify coherent larger regions that these points correspond to. We will add these considerations.*

7.) In some Figures the dpi seems to be too low (e.g. Fig. 3 or 5).

*We will adjust this.*
* * *
**Formalities and Typos:**

• Figure 1 and 8: Change "sahel" and "sahara" to "Sahel" and "Sahara".

• Figure 11 and 12: The meaning of the black dashed lines is not described in the caption.

• Line 34: Change "feed back" to "feedback".

• Line 246: Change "becomes" to "becoming".

• Line 282: Change "daily-change" to "daily change".

*We will revise the manuscript with respect to the Formalities and Typos.*

---

## Author Comment (AC2)

**Reply to Comments of Referee 2** to the Manuscript of Jungandreas et al. "How does the explicit treatment of convection alter the precipitation-soil hydrology interaction in the Holocene African humid period?"

Thank you very much to the very helpful comments!

**General comments**

My main comment is about the presentation of the implications of the results, which are very relevant for the modelling of the GS. The main finding of the paper highlights the importance of considering the precipitation-soil moisture interaction when high resolution simulations are used. I have the impression that this is not sufficiently highlighted in the abstract and in the introduction, when the objective of the paper is stated.

For instance, in the abstract the authors could add a few lines to briefly explain how run-off influences soil moisture instead of referring to a previously published paper. And make a more logical link with the following sentence.

Similarly, in the Introduction, after the description of previous results by Jungandreas et al. (2021), a clear statement is needed of what is missing in the previous paper and what the present paper aims at.

*We will modify and improve the abstract and the introduction.*

**Specific comments**

L14: I don't understand how to connect the conclusions on the role of precipitation type in this sentence with the conclusions on the role of soil moisture in previous sentences. Please make a clearer logical link.

*We will reformulate the last paragraph by mentioning the results found in the simulations with constant soil moisture in the previous paragraph and by more clearly explaining the meaning of the constant-soil moisture simulation. We begin a new paragraph with the sentence starting in line 14, which summarizes not the discussion regarding the simulations with constant soil moisture, but the overall result of our study.*

L82-87: I believe that you should not summarise here your findings. You could outline here the content of the next sections.

*We can reformulate the introduction by shortening the text as suggested.*

L110: why do you use 5 km for simulating explicit convection? Usually in CPM/storm resolving setups, 2-3 km resolution is used. Running the simulations at higher resolution would change your results?

*We had to limit the downscaling simulations to maximum 5 km as the finest horizontal resolution because of computational restrictions. We do not expect qualitatively different results, when going to a finer spatial resolution. First, our simulations at 10km horizontal resolution (using resolved and parameterized convection, respectively) yield not the same, but very similar results. Second, the large mesoscale convective systems of the West African monsoon can be nicely resolved using models with 5km horizontal resolution.*

L130: this sentence is unclear to me. Do you simulate land cover by using IFS or do you use the same land cover used in IFS (as I guess)? Please clarify.

*We checked this again carefully: We use present-day land surface cover generated by the German Weather service (DWD) by combining GLOBCOVER2009, Harmonized world soil database (HWSD), GLOBE & Lake Database. We will revise this in the manuscript explicitly.*

L132-135: I don't understand how you use MPI-ESM Holocene simulations, desert fraction and the present-day vegetation to build your GS land cover. Please rephrase and clarify.

*The present-day land cover is the same as the one used by the land model TERRA, which is used by the DWD for their weather forecast.*

*The MPI-ESM performed 6000 year-long transient Holocene simulations with dynamic vegetation cover. One minus the (dynamic) vegetation fraction gives the desert fraction during the mid-Holocene.*

*We use this desert fraction to know how far north vegetation reached and to determine the main vegetation gradient for our GS-simulations i.e. the lower the desert fraction the higher the vegetation cover. The vegetation gradient we prescribe in our GS-simulation consists of five vegetation types: closed broadleaved evergreen forest , closed to open shrubland, closed to open herbaceous vegetation, sparse vegetation and bare area (desert)(see Fig.2b and d).*

*However, a specific present-day vegetation type cannot always be transformed into the same GS-vegetation type.*
*For example: present-day deserts can be transformed into close broadleaved evergreen forest or sparse vegetation in our mid-Holocene simulations, depending on the latitude or region you are looking at. Therefore, we cannot only use the desert fraction to prescribe mid-Holocene vegetation cover, but also need to specify the present-day vegetation type in each grid cell and its location.*

*We will clarify this part in a revised manuscript.*

L144: which version of ERA? ERA5?

*Yes, we use ERA5. We will mention this explicitly in a revised manuscript.*

L182: in the discussion of the changes in the heat fluxes at the surface, you mention the changes in cloud cover, which are not presented anywhere in the paper. A figure is needed to illustrate cloud cover changes.

*We will add a figure in the Appendix.*

L187: moisture is not limited with regard to what? Do you mean "abundant"? Please clarify.

*Yes, moisture is not limited so that evapotranspiration is not hampered. We will clarify this.*

Section 3.1.2: a map showing the differences in the lon-lat projection would be helpful in illustrating the changes in horizontal transports.

*We will add a map of horizontal moisture transport in the Appendix.*

L243: please add brief descriptions of CAPE and CIN.

*We will add a brief description.*

L246: LFC is not defined.

*We will define it in the revised manuscript.*

Section 3.1.4: this section is not very clear in my opinion. At L252, you state that the increase of vegetation in the coastal region leads to a decrease of precipitation, which I find rather counterintuitive. I'd say that the decrease in precipitation is mostly lead by the change in the thermodynamical gradients due to the dramatic increase in vegetation in the Sahara-Sahel, in turn leading the changes in the dynamics, which you also highlight at the end of the section. Please rephrase to clarify these aspects.

*That is correct, the decrease over the coastal region is mainly caused because of the dynamical shifts of the monsoon further north.*
*We will clarify this in the revised manuscript.*

**Technical corrections**

L75-78: the sentence does not read very well, please revise the location of commas.

L150: We.

L335: please check this sentence, something is wrong/missing.

*The technical corrections will be implemented in the revised version.*

---

## Author Comment (AC3)

**Reply to Comments of Referee 3** to the Manuscript of Jungandreas et al. "How does the explicit treatment of convection alter the precipitation-soil hydrology interaction in the Holocene African humid period?"

Thank you very much for your valuable comments!

**Main comments:**

* At 5 km there is explicit convection but it may be more accurately named convection-permitting rather than explicit convection or storm-resolving.

*There is a long debate in the literature of how to call this type of simulations. We follow the more recent view that what such type of simulations do is to resolve storms, as, as noted by the reviewer, they do not resolve the whole spectrum of convection. In the revised version we will consistently use storm-resolving.*

* One of the major conclusions here is that the intensity of precipitation events is very different between the E and P models. This difference is shown to modify the soil moisture and hence the land-atmosphere feedbacks. One thing that is missing is any analysis of how this intensity distribution of precipitation events actually differs between the E and P models. I realise that it has already been shown in the previous work but it might help to include this here also.

*We will add a similar figure in the Appendix.*

* Related to this, many convection-permitting models overestimate the intensity of rainfall events (e.g. Kendon et al (2021, doi: /10.1098/rsta.2019.0547) because they do not resolve all convection at this resolution. The land-atmosphere feedbacks are dependent on this intensity, so can you comment on this potential caveat? Does ICON overestimate these downpours? More speculatively, would more-fully resolving convection (e.g. down to sub-kilometre scale resolution) address this, or could it sbring to light other effects not considered?

*Figure 6 in Stevens et al. (2020) shows pdf of rain intensity for simulations conducted with a grid spacing of 312 m, 625 m and 2.5 km over the tropical Atlantic and for simulations conducted with a grid spacing of 625 m and 2.8 km over Germany. Over the tropical Atlantic, there is a clear dependency with resolution with much more frequent downpours at coarse resolution, whereas this effect is not present over Germany. Using the same model version as the one used in this study, Paccini (2021, https://pure.mpg.de/rest/items/item_3367420_4/component/file_3367464/content ) investigated the representation of the pdf of rainfall intensity over Amazon, see her Fig. C3. There we can see that ICON with 5-km grid spacing matches very well observations. Hence, although we cannot prove it as the outset, we believe that this effect might be small in ICON. We will add these considerations in the revised version.*

**Minor points:**

Page 7, line 147-156 and point 2: It's not clear what is meant here by the 75th and 85th percentile values? What are these used for? This explanation does not make sense to me.

*We modified the vegetation cover in the GS simulations. To be consistent we also need to modify different parameters in the external parameters that are related to vegetation cover, e.g., surface roughness, LAI, NDVI,… . To prescribe reasonable values for these parameters in the GS simulations, we calculate mean or percentile values from all grid points of present-day vegetation cover of a specific vegetation type. For each parameter, we decided which value - mean, 75$^{th}$ or 85$^{th}$ percentile – are most reliable based on the present-day maps.*
*We will clarify this point in the revised version of the manuscript.*

The fact that the overall precipitation response is not wildly different between the explicit and parametrised convection agrees with studies of future precipitation change in Africa, e.g. Kendon et al (2019, doi: 10.1038/s41467-019-09776-9). It might be worth citing that study here.

*We will cite it in the revised manuscript.*

**Technical corrections:**

Appendix: Please can you add the full descriptions of what each figure shows to the captions in the Appendix figures as well as writing "as in figure x".  Otherwise the reader has to switch between the main text and the Appendix to understand what each figure shows.

Page 7, line 150: "Ee" should be "We".

Table 1: This values would be better at 1 decimal place. Adding an anomaly column GS - DS would be helpful.

Line 204 "Africa thus influences" -> "Africa and thus influences".

Line 366: "Also how the land surface (soil moisture and runoff) reacts to specific precipitation characteristics (drizzle or shower) needs"

-> "How the land surface (soil moisture and runoff) reacts to specific precipitation characteristics (drizzle or shower) also needs"

Line 377: "This study highlights the importance to consider..."

-> "This study highlights the importance of considering...", or similar.

*The technical corrections will be implemented in the revised manuscript.*